# Mid-long term boundary dynamic optimization of open-pit coal mine considering coal price fluctuation

**Shuai Wang, Bo Cao** *, **Runcai Bai, Guangwei Liu**

College of Mining, Liaoning Technical University, Fuxin, Liaoning, China

* caobo0418@163.com

**Data Availability Statement:** All relevant data are within the paper.

**Funding:** The project was supported by the National Natural Science Foundation of China (51974144), the 'Jie Bang Gua Shuai' (Take the

## Abstract

The delineation of the open-pit mining boundary, particularly in the context of medium to long-term planning, forms the foundation of mining design. However, due to the non-linear and dynamic nature of the economic and technical parameters influencing boundary delineation, determining the optimal mining boundary can be exceedingly challenging. Currently, most boundary optimization methods assume that block parameters remain fixed, which results in enterprises assuming a certain level of risk when facing changes in internal and external conditions. In this regard, this paper introduces the concept of "achievement degree" to reflect the risk associated with the results of boundary design. Using coal prices as an example, this article applies the predicted coal price curve to boundary optimization adjustments by specifying the "achievement degree" requirements for various time periods, thereby facilitating risk-controlled and economically optimal boundary decisions. Taking the illustrative case of an idealized small-scale inclined coal seam open-pit mine, adjustments to the boundary closely track variations in coal prices, further enhancing returns. The results demonstrate that the method proposed in this paper can increase overall revenue by approximately 51.15% within the forecast period, while effectively managing risks.

## 1. Introduction

Open-pit mining is a method of extracting mineral deposits by excavating from the Earth's surface. It works by starting at the surface and removing non-economic overlays to access profitable parts [1]. Compared with the underground mining method, open-pit mining has the advantages of high resource recovery, good safety conditions, high labor efficiency and large production scale, and is widely used in the mining of coal and metal deposits [2, 3]. Open-pit mining involves many tasks, including boundary division, engineering organization and schedule design. Among them, boundary delineation is the primary link of open-pit mine design [4, 5]. The essence of open-pit mining production is to extract the economic value behind mineral resources, which requires the deposit in the real world to be expressed as an idealized model with predetermined economic value [6, 7]. Each block has a set of characteristics, including mineral and parting composition, rock type, and density [8]. However, due to

Lead) of the Key Scien-tific and Technological
Project for Liaoning Prov-ince [2021JH1/
10400011], and the Discipline In-novation Team of
Liaoning Technical University (LNTU20TD-07).
There was no additional external funding received
for this study.

**Competing interests:** The authors declare no
conflict of interest.

the constraints of technical conditions and economic considerations, the mining of only a part of geological reserves is generally technically feasible and economically reasonable [9]. Gener-ally speaking, coal mining lasts for a long time, and each stage of the whole life cycle has differ-ent technical, economic and organizational characteristics [10]. The overall economic benefits of open-pit mines are affected by factors such as coal prices and geological conditions [11]. Typically, mining areas contain millions of blocks, and multiple selection decisions need to be made according to actual needs. The existence of uncertainty makes this problem an integrated optimization problem that needs to consider risks [12, 13], which is difficult to be solved with the existing traditional technology [14].

Coal resources are the most important basic energy in China. More than 50% of the energy in electric power, metallurgy, chemical industry, building materials and life industries comes from coal resources [15]. However, a large number of coal resources are continuously devel-oped and utilized, and the mining is not scientific and the utilization is not efficient, which makes the environmental and ecological problems caused by mining emerge in endlessly [16]. The coal industry has long adopted extensive management methods. However, due to the inherent scarcity and limitation of resources, the problems of resources and resource utiliza-tion are prominent [17]. China 's coal resources reserves are in the forefront of the world, but the per capita share is low. If a complete legal system for mining and mining management is not established, it will seriously affect China 's energy management [18]. The mining right sys-tem is the core content of the legal system of mineral resources. The mining right clarifies the scope of mining allowed by law in open-pit coal mines and is a rigid constraint that must be observed in the process of delimitation. The complexity and rigor of mining laws and regula-tions may have a wide impact on the feasibility and design of open-pit mines. Mining compa-nies must closely comply with legal and regulatory requirements when designing and implementing projects to ensure project feasibility and sustainability while meeting environ-mental and social expectations.

The activities of mining companies are not only sensitive to market factors (price fluctua-tions, demand changes), but also sensitive to non-economic global challenges (climate change, regional conflicts). They must cope with the decline in the quality and availability of mineral resources and the increase in the amount of mine waste [19]. Therefore, in the past few years, the issue of sustainable development (SD) has become the subject of many discussions [20]. Mining can have strong negative impacts on the natural environment in the form of soil dis-turbance, waste rock disposal, water pollution and drainage, landscape destruction, harmful discharges, water contamination by industrial effluents, flora and fauna degradation and other negative impacts on ecosystems [21]. Based on this, the concept of circularity has been gradu-ally introduced into mining [22]. Circular economy describes production in a circular model where markets, regulations, and industrial systems are optimized to design high-performance products, minimize impact, restore or regenerate the environment, and optimize material use [23]. And the mining circular economy refers to the economic system that follows the charac-teristics and natural ecological laws of mineral resources and mineral products, and takes the efficient development and comprehensive utilization of mineral resources as the core [24]. Most scholars agree that the principle of circular economy should be followed is 'Reduce, Reuse, Recycle', referred to as '3R'. In the process of mining, processing and utilization of min-eral resources, the circular economy is mainly manifested as: (1) through mechanization, auto-mation and optimization of mining, to achieve efficient exploitation of resources; (2) By studying the mining and smelting technology of complex and difficult-to-mine and difficult-to-separate ores, the mining dilution rate and ore loss rate are reduced, the recovery rate of beneficiation and smelting is improved, and the total recovery rate of resources is improved. (3) Reduce the discharge of various pollutants such as tailings, coal gangue and mine

wastewater, and improve the comprehensive benefits of resource development [25]. Circular economy is one of the key directions of mineral resources conservation and protection of sustainable development policy [26]. For emerging economies, including Brazil, Chile, China and South Africa, their economic dependence on mining and extractive industries is relatively large, so the research of circular economy is very important [27].

With the help of the '3R' principle of circular economy, optimizing the mining strategy of coal mines can better consider resource efficiency, environmental sustainability and economic benefits. Changes in economic indicators, especially coal prices and production costs, are usually the main factors affecting the economics of open-pit mining schemes. However, in general, unless the mining process changes significantly, the production cost will fluctuate within a certain range, so the impact of coal prices on the economy of open-pit mines is very important. Accurate forecasting results of coal prices can help to equationte more effective management policies and obtain greater benefits [28]. The basic principle of coal price prediction can be understood as, collecting past coal price data, through the analysis of its law, to obtain the change of coal prices in the future, which can be solved by statistical economics. Jiang et al. [29] estimated China's coal prices, consumption, and investment from 2016 to 2030 using the ARIMA model. Lasheras et al. [30] predicted the copper price trend with the help of a neural network and ARIMA model. Duan [31] predicted the price of gold futures based on time series analysis. Zhang [32], Hu [33], and Gu [34] carried out extensive research on time series prediction by means of econometrics.

However, because uncertainty is an inherent property of coal prices, risks are always present in the process of predicting coal prices and guiding production [35]. Therefore, it is necessary to find a suitable measurement method for the risk of coal price. At present, the two mainstream risk measurement methods in the energy field include CVAR and entropy risk measurement [36]. Based on this, once the risk is effectively measured, the optimal mining decision including risk can be made in an uncertain environment by controlling the risk [37, 38]. Therefore, based on the actual production of open-pit coal mine, the author tries to put forward a method to optimize the mining boundary of open-pit mine by considering the change of coal price under the condition of fully considering the constraints of mining right and law. The ultimate goal is to improve the net present value in a specific period of time under the condition of risk control. Considering the difference between the near horizontal coal seam open-pit mine and the inclined coal seam open-pit mine in the boundary division work, the optimization effect evaluation index under the unified research standard is proposed, which is called "achievement degree". The goal of the study is to propose a dynamic adjustment strategy that comprehensively considers constraints including policy, economy, and mining conditions, and is suitable for the actual production of open-pit coal mines. The specific completion includes predicting coal prices based on historical data, constructing a unified optimization effect evaluation standard, and proposing a multi-step dynamic optimization method. Finally, through the empirical analysis of the inclined coal seam open-pit mine, the feasibility and effectiveness of the proposed method are verified. Practice has proved that by considering the fluctuation of coal price, the net present value can be increased by about 50% in the prediction period. Therefore, the method proposed in this article is feasible and effective in dealing with the dynamic optimization problem of the open-pit boundary.

## 2. Boundary dynamic optimization method considering uncertainty

A basic problem in open-pit mine planning is to delineate the mining range of the mine, and its size affects the final amount of mineable ore and stripped rock, as well as the production scale and mining sequence. This problem usually expresses a mine as a series of blocks, with

the goal of maximizing the net present value, by changing specific indicators, generating a certain number of nested pits, deriving different stages of the mine mining sequence, and then proceeding to production Scheduling optimization.

Nested pits are widely used in the optimization of open-pit mines. The basic idea is to solve the open-pit mine optimization problem under this condition given a series of non-negative income factors $\beta_1 < \beta_2 < \ldots < \beta_k$. The solution of the problem satisfies the nested coverage, which can provide effective guidance for optimizing the boundary. However, the effectiveness of nest method has been questioned: (1) When the economic situation is good, some coal resources can-not be extracted in a timely manner, and then part of the benefits may be lost; (2) When the economic situation is bad, some resources in the original realm may be lost. Mining this portion of coal resources may result in additional losses. Therefore, it is necessary to improve the three-dimensional block nesting method, incorporate uncertainty into the decision-making process, and effectively deal with risks when the economic form changes.

## 2.1 Improvement of three-dimensional block nesting method

As a method of deposit representation using a 3-D array of block elements, the block model has good adaptability to solve this problem. The blocks mined within the boundary generally only include two destinations: the stripped material is transported to the dump for disposal through the transportation line, and the raw coal enters the mine's main transportation system and is transported to the coal storage yard for further processing.

Uncertainty runs through every link in the whole process from the collection of open-pit deposit information to the delineation of the open-pit boundary. The existence of uncertain factors leads to strong instability in the final determined boundary. Therefore, it is necessary to effectively measure and target the uncertain factors Improve the design method to achieve the purpose of controlling the risk of the whole process. Based on the three-dimensional massive deposit model theory, starting from the integer programming method, the open-pit boundary design process is shown in Fig 1.

For all blocks within the mining scope, the mining cost $c_b$ is assigned. In addition, given the priority constraints of each block, follow the maximum slope angle constraint of the edge. To simplify the explanation process, let $p_b$ be a random variable representing the profit of the block. Assuming that the mining unit volume block cost is $c_b^e$, the unit transportation cost is $c_b^t$, the unit post processing cost of the ore block is $c_b^p$, the inner and outer dischargee distances of

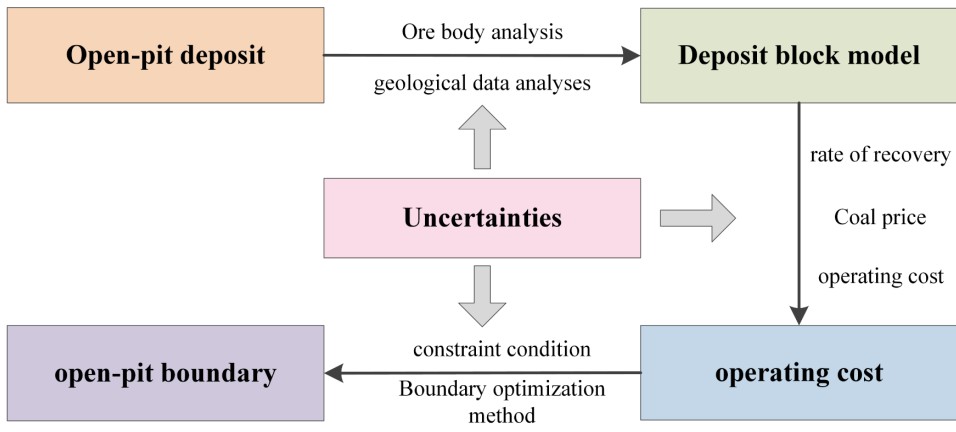

**Fig 1. The flow chart of open-pit mine optimal boundary design.**

the stripping material are $L_n$ and $L_w$, respectively, and the raw coal transportation distance is $L_m$. The cost $v_b$ of the extracted block can be calculated by the following equation:

$$v_{b=} \begin{cases} c_b^e + c_b^t \times L_m + (c_b^p - r_b p_b) (\text{ore block}) \\ c_b^e + c_b^t \times L_n (\text{Internal waste block}) \\ c_b^e + c_b^t \times L_w (\text{Exhaust waste block}) \end{cases} \quad (1)$$

In the equation: $p_b$ is the treated unit block profit; $r_b > 0$ is used to compensate for other factors that affect block profits.

Let the $P \subseteq B \times B$ to be a priority constraint set, that is, for $(b,b') \in P$, while block $b$ is mined, it is necessary to ensure that block $b'$ has been mined, and the medium and long-term boundary optimization problem is expressed by cost minimization:

$$UP = \min_{x^e, x^p} \sum_{b \in B} (c_b^e x_b^e + c_b^t x_b^n \times L_n + c_b^t x_b^w \times L_w + c_b^t \times L_m) + \sum_{b \in B} (c_b^p - r_b p_b) x_b^p$$

$s.t.$

$$x_b^e \le x_{b'}^e, \forall (b, b) \in P, \quad (2)$$

$$x_b^e, x_b^p, x_b^n, x_b^w \in \{0, 1\}, \forall b \in B,$$

Among them, $x_b^e \le x_{b'}^e$ is the block priority constraint; $x_b^p$ is the post-management decision; $x_b^e$ is the mining decision; $x_b^n$ is the inner drainage decision; $x_b^w$ is the outflow decision; $x_b^p, x_b^e, x_b^n$ and $x_b^w$ are all binary random variables with values of 0 or 1.

## 2.2 Selection of risk measurement methods

The fact that the parameters used to determine the block economic value are fixed and the influence of uncertain elements is not taken into consideration is one of the drawbacks of the classic border optimization method. It is vital to specify the ideal properties of the mine produced by the risk measure when the level of risk aversion is modified in order to choose an appropriate risk measure for the aforementioned issues. The nested notion can be used to create a collection of nested mines with consistent risk levels under the assumption of risk management.

The risk nesting characteristic in the process of open-air optimization design can be derived by fusing the two requirements of constant normalization and translation consistency in risk measurement [6].

(1) Risk nesting

The main results are as follows: (1) the risk nesting assumes that the blocks remain independently distributed in the mining area, and when the parameter $\alpha$ increases, the risk measure $\rho_\alpha$ of the decision-making process increases. The pit generated by a lower level of risk aversion $\alpha_1$ should be completely included in the pit generated by a higher level of risk aversion $\alpha_2$.

(2) Additive consistency

Assumes that $X$, $Y$, and $Z$ are random variables and $Z$ is independent of $X$ and $Y$, and the variable $\rho_\alpha(\cdot)$ represents the risk measure under the degree of risk aversion $\alpha$. If the risk aversion degree $\alpha$ $\rho_{\alpha(X)} < \rho_{\alpha(Y)} \to \rho_{\alpha(X+Z)} < \rho_{\alpha(Y+Z)}$ is different, then $\rho_\alpha(\cdot)$ satisfies additive consistency. That is, adding a certain number of blocks in different stages of the mine can not change its original risk measurement.

Usually, the risk of an open-pit mine includes the loss of profit and the decrease of profit that may occur in an open-pit mine, which is closely related to the fluctuation of profit index including production conditions and market. Risk measurement means that the uncertainty

that may be generated is expressed in a specific numerical form, that is, the size of the risk measurement is expressed by a specific numerical value. If $X$ is a random variable, then the entropy risk measure of $X$ under the risk aversion coefficient $\alpha \in [-1,1]$ is defined as:

$$
\rho_\alpha^{Ent}(X) = \begin{cases} \dfrac{1}{\alpha} logE[e^{\alpha X}] & \alpha \neq 0, \\ E[X] & \alpha = 0. \end{cases} \tag{3}
$$

When $\alpha \rightarrow 0$, $\rho_\alpha^{Ent}(X) = E[X]$ is neutral (deterministic); when $\alpha \rightarrow 1$, $\rho_1^{Ent}(X) = ess\ supX$, the decision maker tends to avoid risk; when the risk is 1, the decision maker tends to take the risk. When $\alpha \rightarrow -1$, $\rho_{-1}^{Ent}(X) = ess\ infX$, the decision maker tends to take the risk. Entropy risk is a series of convex but discontinuous risk measures, which reflects the degree of uncertainty, satisfies the additive consistency, and has good conditions for application to the optimization problem of the open-pit mine.

The production plan in accordance with the original design boundary line is called into question when the production of an open-pit mine is impacted by unpredictable circumstances. The open-pit mine tends to bear risk when $\alpha < 0$, so it is necessary to expand the open-pit mine boundary and reduce the enterprise's loss profit; when $\alpha > 0$, it tends to avoid risk, so it is necessary to reduce the open-pit mine realm and reduce the enterprise's loss profit. This is how the entropy risk definition of Eq (3) can be used to reflect the different risks faced by the open-pit mine.

In entropy risk measurement, the selection of risk aversion coefficient α is very important. If $\alpha$ is positive, it will increase the weight of high probability events in the risk measure, thereby increasing the overall risk. If $\alpha$ is negative, it will reduce the weight of high probability events, thereby reducing the overall risk. The determination of $\alpha$ value usually needs to consider the specific decision-making situation and the preference of decision-makers. Based on the sensitivity analysis method to determine the sensitivity of α value to decision-making, the appropriate value is selected according to the historical data information and the preference of decision-makers. However, the decision-making problem and the preference of decision makers may change with time, and the $\alpha$ value needs to be updated and corrected according to the actual situation.

It is worth noting that due to the long-term extensive mining mode adopted by the coal industry, coupled with the large-scale mining and utilization of coal in recent years, mineral resources have been depleted. Because the initial planning and design tends to have better mining conditions and shallower shallow coal resources, the conditions of coal resources in the subsequent development process become worse. For example, when the boundary of open-pit mining in inclined coal seam is delineated, in order to ensure the economy of its boundary, it is usually considered that when the mining depth exceeds a certain range, the deep resources are mined by underground mining. However, it should be noted that the coal resources adjacent to the open-pit mine have a natural disadvantage in terms of safety compared with the original open-pit mining method. Therefore, it is theoretically feasible to expand the original open-pit mining boundary under the conditions allowed. Therefore, in the process of determining the risk aversion coefficient, it is possible to have a tendency to bear certain risks, thereby increasing the reserves of open-pit mine resources, extending the service life and obtaining more resources as much as possible, and making beneficial contributions to the management of mining resources. Of course, in this process, one of the factors that must be considered is the legal factors and environmental protection restrictions.

## 2.3 Two-stage decision-making optimization method considering uncertainty

Considering the construction and development procedure of open-pit mine, the multi-step boundary optimization method of an open-pit mine under the condition of risk control is given. That is, a certain stage in the development and construction of the open-pit mine is designated as stage "0", and a series of technically feasible alternative nested pits are obtained by giving different risk levels, which are defined as stage "1". Considering that the open-pit mine cannot make a large-scale adjustment to the length of the working line in a short period of time, the adjustment of the length of the working line should not exceed 20% of the current length of the working line.

Due to the great uncertainty of the economic index of the block in the subsequent stage of mining, the overall economic benefit of the mine can not be reflected in a single stage. It is decided to use the same idea in stage "1" to generate a series of alternative nested pits, to form the alternative decision space of stage "2", and to select the optimal mining boundary of stage "1" with the goal of maximizing the total net present value of the two stages. By analogy, the optimal mining scheme for all stages of the optimization period is determined and only determined, as shown in Fig 2.

In Fig 2, assuming that the mining boundary of the open-pit mine at stage "0" is $S_0$ and using 20% boundary adjustment as the interval, the alternative mining schemes for stage "1" can be generated in the order from small to large as $s_1^{0.8}$, $s_1^1$ and $s_1^{1.2}$ ($s_1^1$ means that the length of the working line has not changed, $s_1^{0.8}$ means that the length of the working line is adjusted to 0.8 times that of the original scheme, and the number of alternatives can be increased by changing the adjustment interval. In the same way, phase "1" generates phase "2" alternatives in the same way, generating a total of 9 different realm combinations.

Among them: $s_1^{1.2}$ and $s_2^{1.2}$ constitute the scheme with the largest boundary size, which is called the maximum boundary of two-stage size; $s_1^{0.8}$ and $s_2^{0.8}$ constitute the scheme with the smallest boundary size in the scheme, which is called the minimum realm of two-stage size; in all schemes, there is always a realm with the highest degree of fit with the reality, which can better reflect the volatility of the factors affecting profits, which is called the highest state of achievement. In this example, it is assumed that $s_1^1$ and $s_2^{1.2}$ constitute the highest degree of achievement.

In order to solve the problem of medium-and long-term boundary optimization in open-pit mines, a series of nested pits are generated according to the isometric sequence. Suppose the block profit $p_b \sim N(\mu, \Sigma)$ in each ore block, where $\mu$ represents the average vector and $\Sigma$ is the variance-covariance matrix. In this case, the medium and long-term boundary optimization problem using entropy risk measurement can be expressed as the following mixed integer programming problem:

$$\min_{x^e, x^p} \sum_{b \in B} (c_b^e x_b^e + c_b^t x_b^n \times L_n + c_b^t x_b^w \times L_w + c_b^t \times L_m) + \sum_{b \in B} (c_b^p - r_b p_b) x_b^p$$
$$+ \frac{1}{2} \alpha \sum_{b \in B} \sum_{b \in B} r_b x_b^p r_b x_b^p \sum_{bb} \tag{4}$$

Based on the principle of cost minimization and combined with scenario planning, Eq (4) can generate a series of nested pits at a specific time node at a given risk level, and obtain and only get a set of optimal nested pits with the goal of minimizing risk. considering risk minimization, multi-stage nested pits are obtained, taking into account both economic benefits and enterprise risks.

**Fig 2. Sliding window method to determine optimal mining boundary.**

## 3. Construction of dynamic optimization evaluation system

### 3.1 Analysis of boundary adjustment strategy based on whole life cycle research

Combined with the characteristics of open-pit mine production, it can be seen that in the whole life cycle, the production capacity first has a gradual improvement process, and then the output gradually returns to zero in the end period, completing the mining of all ore bodies in the boundary. Take an open-pit coal mine as an example, from the construction of the mine to the end of mining, to go through the construction period, production, delivery, stripping peak, and the end of the production reduction period. Based on this, the life cycle of an open-pit

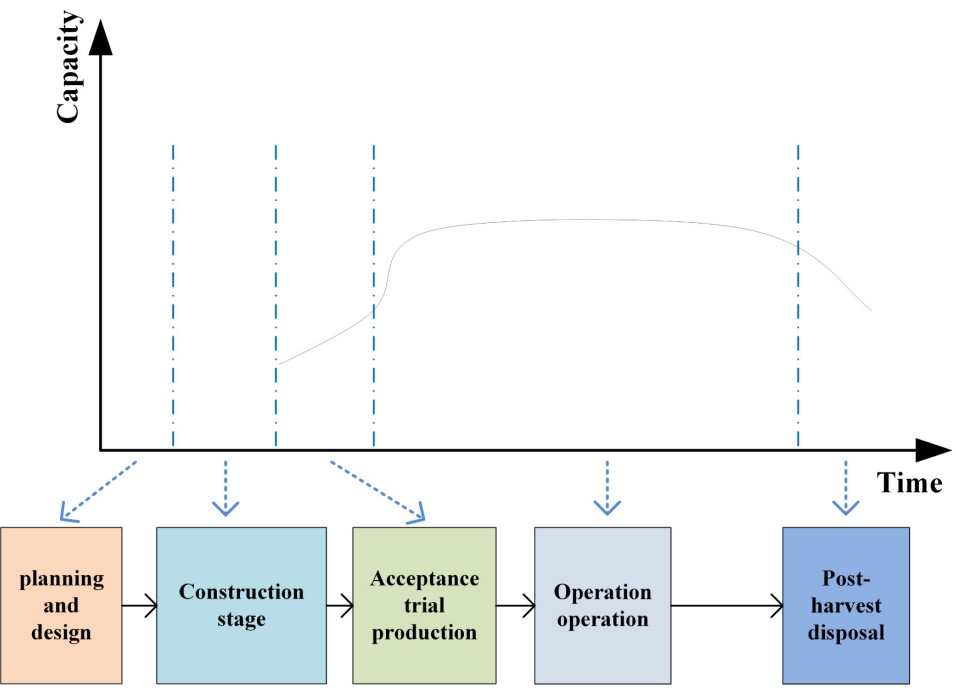

**Fig 3. Full life cycle stage classification of open-pit mine.**

mine is divided into stages such as planning and design, construction, acceptance trial production, production and operation, and post-mining disposal, as shown in Fig 3.

According to the size of resource reserves and annual production capacity, open-pit mines can be divided into small, medium and large open-pit coal mines. Generally speaking, the final mining boundary of large and medium-sized open-pit mines is very wide. In order to ensure the continuity of the project, the final mining boundary will be divided into several smaller mining areas by stages or partitions. Due to the small reserves of resources, small open-pit coal mines are generally divided into only one mining area for mining. As mentioned above, the mining boundary is the basic document for the preparation of the mining plan. The mining plan can be divided into overall planning, long-term planning, and annual planning. Therefore, according to this standard, the boundary is divided into three types: full mining boundary, medium and long-term boundary, and short-term boundary. Combined with the actual production of open-pit mine and the differences in coal seam occurrence, the adjustment of open-pit limit in near horizontal and inclined coal seams is described respectively. The schematic diagram of different types of open-pit limit adjustment in the section of the end side is shown in Fig 4.

Limited by mining engineering, the boundary of open-pit mine can not be adjusted in a short time, so the dynamic optimization scheme of the boundary is only for the final mining boundary and the medium and long term boundary.

As shown in Fig 4(A), the adjustment of the open-pit mining boundary of the near-horizontal coal seam can be approximately regarded as caused by the outward expansion or inward contraction of the surface boundary. The final mining boundary is generally the mining right boundary, which can only be adjusted when there are major changes in the internal and external environment. The boundary line of the partition boundary can change with the change of economic indicators, so the research object of the dynamic adjustment of the boundary is only for the partition boundary.

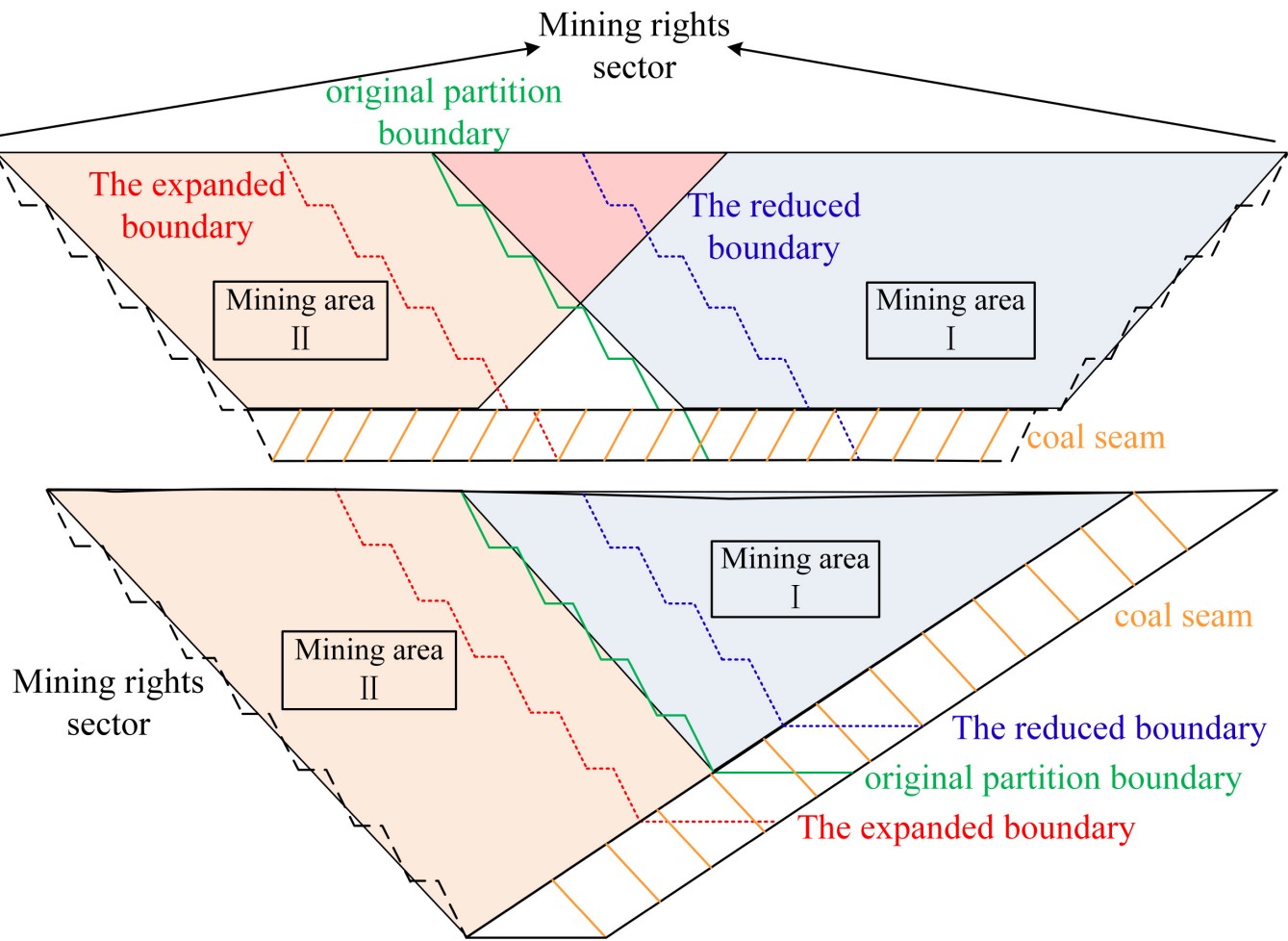

**Fig 4. Schematic diagram of boundary adjustment section of open pit mines.** (a)Open pit limit adjustment of the near level coal seam. (b)Adjustment of open pit limit in inclined coal seam.

As shown in Fig 4(B), due to the inclined coal seam in open-pit mine, the change of economic index affects the economy of coal resource mining in the original boundary, which leads to the change of optimal mining depth. Therefore, for the inclined coal seam open-pit mine, the dynamic adjustment of the boundary is not only applicable to the partition boundary, but also to the final mining boundary.

In the dynamic process of realm, two goals need to be considered. On the one hand, the mining company can obtain the maximum profit after the block mining, on the other hand, it can meet the various constraints affecting the feasibility of the boundary and ensure the minimum risk. At this time, how to reasonably control the risk under the unified evaluation system and ensure the maximization of mine revenue has become the focus of research. Therefore, it is urgent to develop an evaluation system for the production characteristics of open-pit coal mines.

## 3.2 Construction and application of achievement evaluation system

For a long time, the aim of open-pit mining has focused on maximizing profits, and the stripping ratio is the only indicator to determine the construction plan of open-pit mines, while

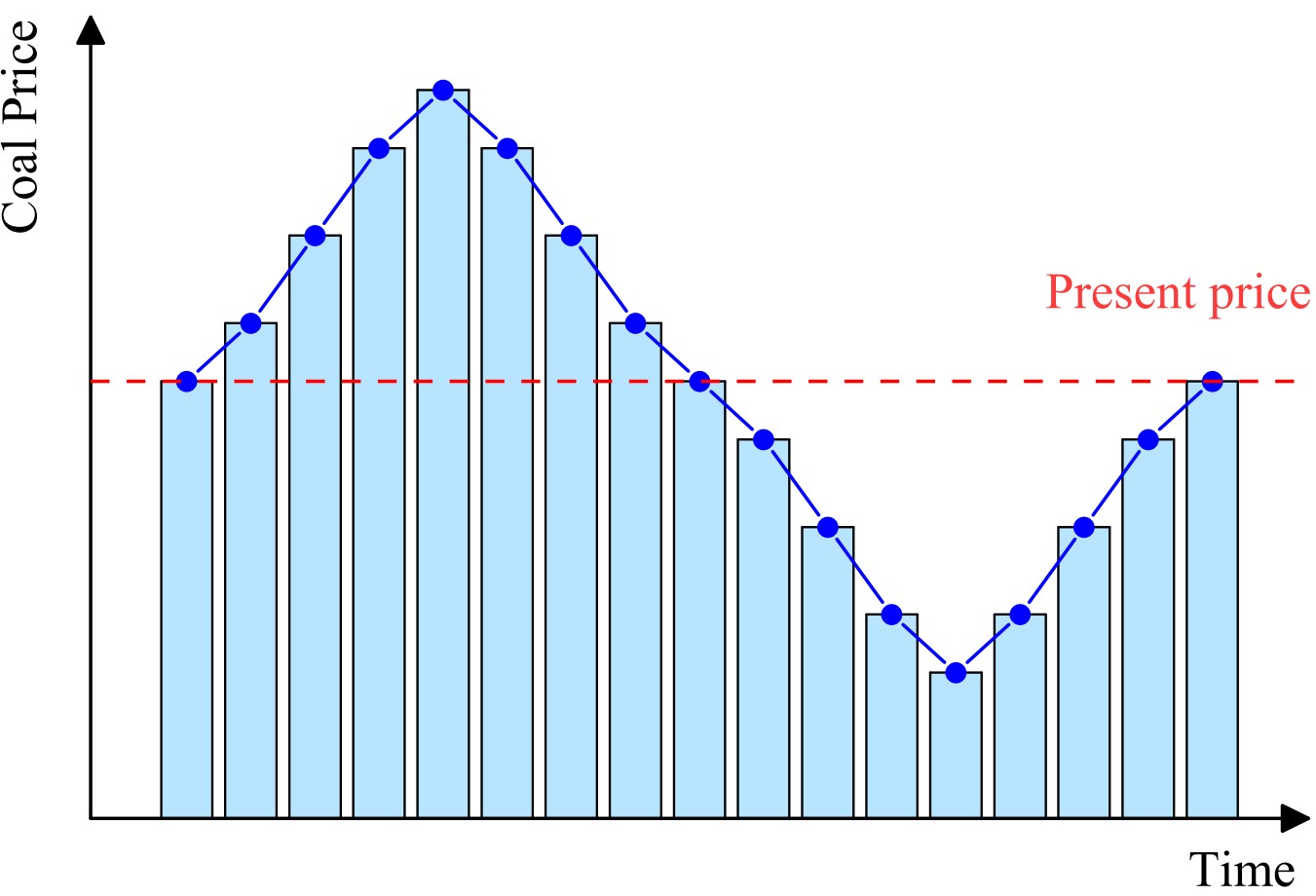

**Fig 5. Comparison chart of actual coal price and fixed coal price index.**

largely ignoring the risks faced by enterprises. There is always a certain deviation between the coal price index used in the initial design and the actual operation: when the actual price is higher than the estimated price, the company will suffer indirect losses because it cannot sell a large amount of coal at a high level in time; When the actual coal price is lower, the extraction of a part of coal may mean a loss.

As shown in Fig 5, it is assumed that the price of raw coal always fluctuates, which constitutes a deviation from the preset coal price: (1) When a positive deviation occurs, the enterprise will increase part of its income due to the increase in coal price, but at the same time, due to the boundary limit It is impossible to increase coal output to further increase profits, so it can be considered to expand the open-pit mine boundary within the time range when the deviation may occur; (2) When the negative deviation occurs, the enterprise will cause its own losses due to the reduction of coal prices, but at the same time because Boundary restrictions cannot reduce coal output and further increase losses. Consider reducing the open-pit mine boundary within this time frame in order to effectively deal with the risks brought about by uncertainty.

With reference to the robustness problem in economics, the concept of "achievement degree of open-pit mine optimization design" is proposed. The "achievement degree" at a specific time represents the correlation coefficient between the change rate of economic indicators and the change rate of boundary adjustment indicators, which is used to test the degree of fitting between the optimization results and the predetermined demand, and is expressed by the

symbol $c_s$. Its value range is usually between −1 and 1, −1 indicates a completely negative correlation, 1 indicates a completely positive correlation, and 0 indicates no linear correlation. Its calculation formula is $c_s = \left(1 - \frac{|CR-LR|}{(CR+LR)/2}\right) \times 100\%$, where, *CR*-economic index change rate, *LR*-boundary adjustment rate.

In order to simplify the description process, it is assumed that the change rate of boundary adjustment at a certain stage is related to the length of the working line at that stage during the development of the open-pit mine. Assuming that the production capacity can be dynamically adjusted, the greater the production capacity, the longer the length of the working line at this stage under the same propulsion speed. The "achievement degree" reflects the relationship between the length of the working line and the deviation in the optimization process. The greater the degree of achievement $c_s$, the more sensitive the enterprise is to capture the uncertain factors, which is equivalent to the greater profit of the open pit mine at the present stage.

Medium and long-term boundary optimization can effectively deal with the risks caused by uncertainties in the whole life cycle. In the stage of planning and design, use fixed coal price indicators to determine the final mining state under the current conditions; in the stages of construction, acceptance trial production, production, and operation, etc., with changes in internal and external environments, based on the classification of the whole life cycle stage, After the uncertainty is gradually exposed, the recourse decision is made considering the different "achievement degree" requirements of each stage, and the boundary is adjusted in time, in order to effectively deal with the complex uncertainty of the production process.

There is a corresponding relationship between the design achievement requirement and the corresponding risk of the realm, that is, $c_s + |\alpha| = 1$. There are differences in the achievement degree in each period of open-pit mine construction and development, but it cannot be less than the required minimum achievement degree. Use the deviation between the actual price and the preset price to indirectly calculate the attainment degree $c_s$ of the boundary optimization design within the entire life cycle to reflect the advantages and disadvantages of the boundary adjustment plan; after the acceptance and trial production of the open-pit mine is completed, the minimum achievement requirement $c_{smin}$ for each period is indirectly calculated by giving different risk levels, and the adjustment of the boundary is standardized.

For example, if the actual price is 20% lower than the preset price in a certain period, the design achievement $c_s$ = 80%; at the same time, enterprises tend to avoid risks during this period, given the risk aversion level $\alpha$ = 0.1, the minimum attainment degree $c_{smin}$ = 0.9>0.8. At this time, the minimum achievement requirement cannot be met, and production needs to be carried out in accordance with the regenerated open-pit mine realm after adjusting the economic indicators to meet the minimum achievement requirement and effectively deal with risks.

## 4. Data collection and processing

### 4.1 Analysis of coal price prediction method

**4.1.1 Construction of coal price forecasting model.** As mentioned above, raw coal price has the most far-reaching impact on the realm, so this paper takes the uncertainty of coal price as an example to illustrate the influence of uncertain factors on the boundary optimization design of open-pit mines, and the research results can be extended to other types of uncertainty. Collect raw coal prices and thermal coal prices in the decade from 2010 to 2020 [32]. The trend comparison is shown in Fig 6.

As seen in Fig 6, there is a significant correlation between the price of thermal coal and the price of raw coal pits, meaning that both have essentially the same shifting trends. The pit

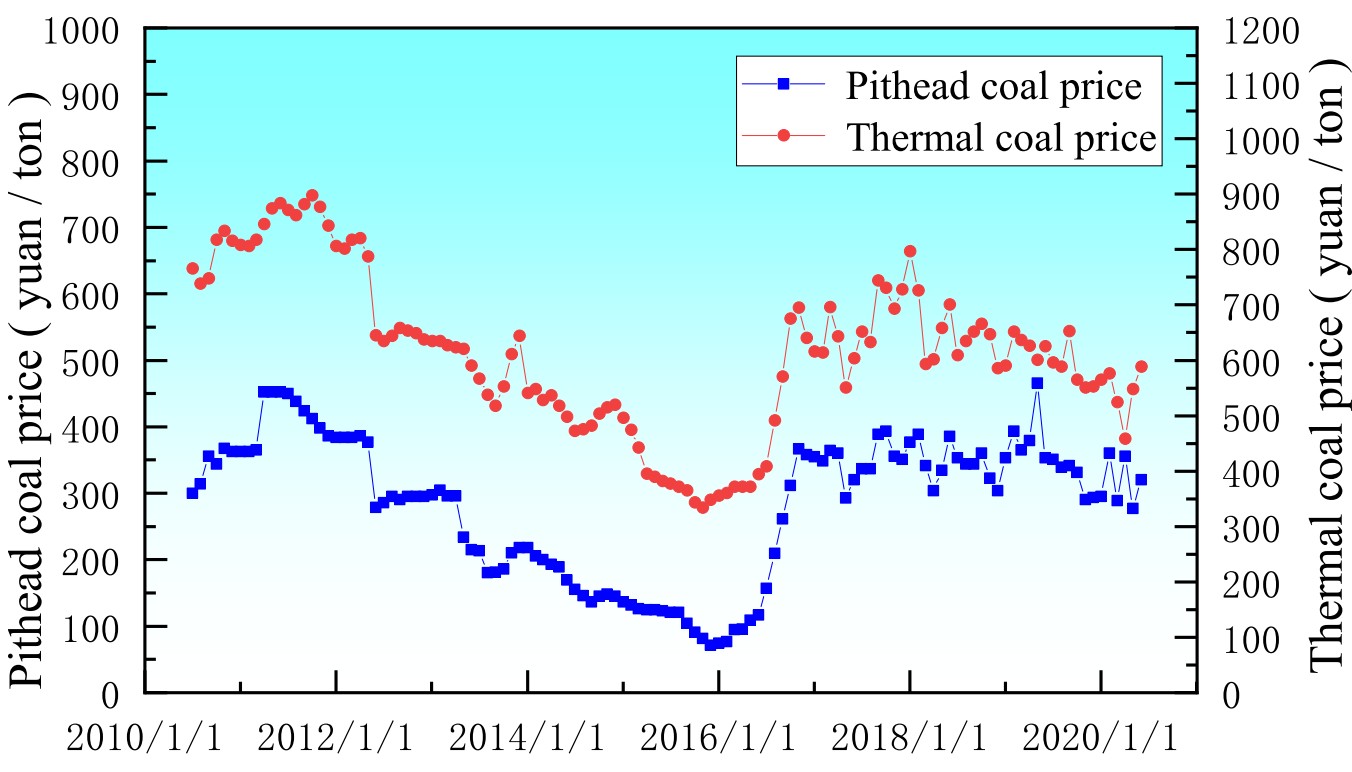

**Fig 6. Comparison of raw coal pit price and thermal coal price trend.**

mouth price of coal is used as the primary research object in this work since the open-pit mine directly sells raw coal, which is how the business derives its eco-nomic worth. The historical pattern demonstrates that the price of unprocessed coal pit has fluctuated frequently and sharply.

**4.1.2 Coal price prediction model construction and data preprocessing.** In recent years, the price time series are frequently predicted using the autoregressive moving average model (ARIMA). Three indexes—the autoregressive term $p$, moving average term $q$, and difference number $d$—control the model primarily. In other words, the ARIMA model has the following form:

$$\varphi(L)(1-L)^d X_t = \theta(L)\varepsilon_t \tag{5}$$

In the equation: $\varphi(L)$ and $\theta(L)$ are stationary lag operators of order $p$ and $q$, respectively; $L$ is a lag operator, that is, $LY_t = Y_{t-1}$; $\{\varepsilon_t\}$ is a white noise sequence; $d$ is a difference parameter; $c$ is a constant; $p$, $d$, $q$ are all non-negative integers.

According to time as the standard of price measurement, it can be divided into the following categories: long-term price forecasts of five years and more, medium-term price forecasts of one to five years, and short-term price forecasts based on the planned year. As mentioned earlier, the boundary adjustment of the open-pit mine needs a certain reaction time, that is, the adjustment of the boundary always lags behind the change of coal price, so it is reasonable to forecast the medium-term coal price.

Choosing the daily raw coal price to calculate the monthly average price can more effectively reduce the price forecast inaccuracy by collecting the raw coal pit price data from July 2010 to June 2020. The data set is broken up into 120 monthly groups of coal price data in total, with the first 100 groups serving as training sets and the final 20 groups serving as

verification sets. The verification set is used to test and update the model's parameters after the training set has been used to train the models.

In general, the price series predicted by the ARIMA model is divided into seven steps: (1) import historical coal price data to generate time series for stationarity test. (2) if it does not pass the stationarity test, the time series is con-verted into a stationary time series by difference processing. (3) White noise test, if the sequence is white noise sequence, it does not have the feasibility of analysis; if it is non-white noise sequence, modeling and prediction. (4) the model is identified, and the corresponding model is determined according to the trailing and truncating characteristics of autocorrelation and partial autocorrelation graph. (5) calculate the parameters of the model. (6) the feasibility of the model is verified, and the white noise test is carried out on the residual sequence. if it is a white noise series, the model is effective, otherwise it is invalid. (7) the calculated model is used to predict the future coal price series.

Since the fixed coal price index cannot be adjusted to account for changes in coal price, it was chosen as the basis for drawing the original border of an open-pit mine. The enterprise typically bears more risk the larger the discrepancy between the predetermined coal price index and the actual coal price data. Based on the predicted coal price and the preset achievement demand, with the help of the dynamic optimization method of the boundary proposed in the previous section, the dynamic optimization of the boundary can be carried out to improve the economic benefit under the given risk (controlled by the minimum achievement degree).

## 4.2 Topological structure of a small open-pit mine with inclined coal seam

As shown in Fig 4, the boundary adjustment of inclined coal seam open-pit mine not only involves the adjustment of partition and stage boundary, but also includes the final mining boundary change caused by the change of mining depth. Therefore, it is more representative and convincing to choose inclined coal seam open-pit mine as an example. Thus, an idealized small open-pit coal mine is chosen as the research object in order to demonstrate the usefulness of the method in this work in solving the boundary optimization problem of open-pit mine under unknown conditions. The opencast coal mine's progress is expected to be consistent every year in order to ensure the model's effectiveness. Fig 7 depicts the small inclined coal seam open-pit mine's two-dimensional topology.

It is presumptive that the coal seam's inclination angle and the final slope angle are both 45˚. At the same time, the priority constraint is derived from the given block and the exfoliation of the inner and outer discharge is distinguished. The main economic indicators of the production link of the open-pit mine are given as shown in Table 1. It is assumed that the final mining boundary of the open-pit mine is flexible when the external environment changes, i.e., when the coal price exceeds the preset value, the mining depth can be appropriately increased, the open-pit mine boundary and the output of raw coal can be expanded, and vice versa.

## 4.3 Optimal design of open-pit mine boundary under coal price fluctuation

The service life of open-pit mines is generally long, and the coal price fluctuates in various periods. The changes in coal prices in the coming period include three trends: rising, falling and basically stable. The simplest change trend of combined coal price can be obtained by the pairwise combination of three kinds of coal price change trends. The change of open-pit mine boundary in a particular period can be divided into five types: contraction, expansion, contraction-expansion, expansion-contraction and basic stability, and the corresponding boundary trend is shown in Fig 8.

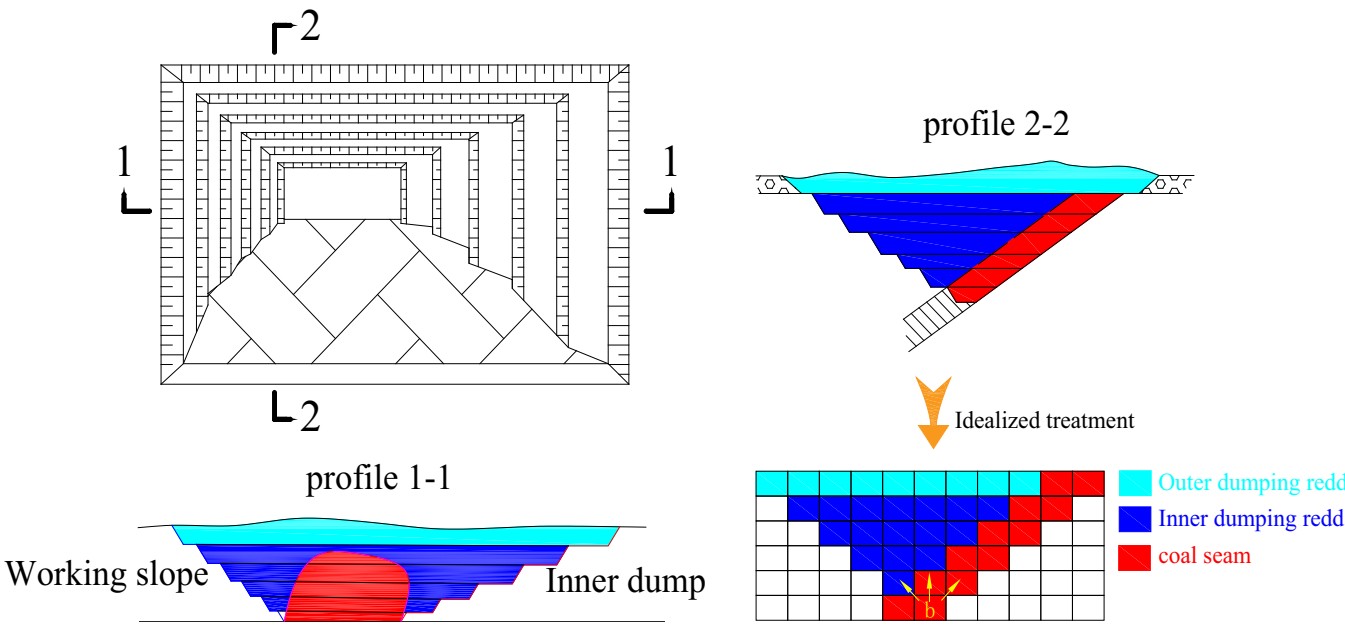

**Fig 7. Structure of a small inclined coal seam open pit mine's two-dimensional topology.**

Fig 8(A)–8(E) shows five state trends of contraction, expansion, contraction-expansion, expansion-contraction and basic stability, respectively, and the corresponding changes in coal prices are shown by coordinate diagrams. In the whole life cycle of open-pit mine, the changing trend of coal price is more complex, including the different permutations and combinations of the above five changes.

The various stages of the entire life cycle of the open-pit mine were used as the starting point for the illustration process, along with the multi-step boundary optimization method shown in Fig 2. This method was combined with the multi-step boundary optimization method to simplify the illustration process, guided by the demand of reaching degree in each stage, along with the actual production demand of the open-pit mine, taking into consideration the changing trend of coal price.

## 4.4 Optimal design of open-pit mine boundary under coal price fluctuation

When the price of raw coal is greater than the predetermined value, the risk aversion coefficient is negative; when the price of raw coal is lower than the predetermined value, the risk

**Table 1. Principal economic indicators in the production chain.**

| Index | Unit | Value |
|---|---|---|
| Mining cost | yuan/m$^3$ | 10 |
| Transportation costs | yuan/m$^3$/km | 2 |
| Post-processing cost | yuan/m$^3$ | 5 |
| Internal discharge distance | km | 2 |
| External discharge distance | km | 4 |
| Raw coal transport distance | km | 3 |
| Preset coal profit | yuan/m$^3$ | -60 |
| Resource recovery rate | % | 95 |

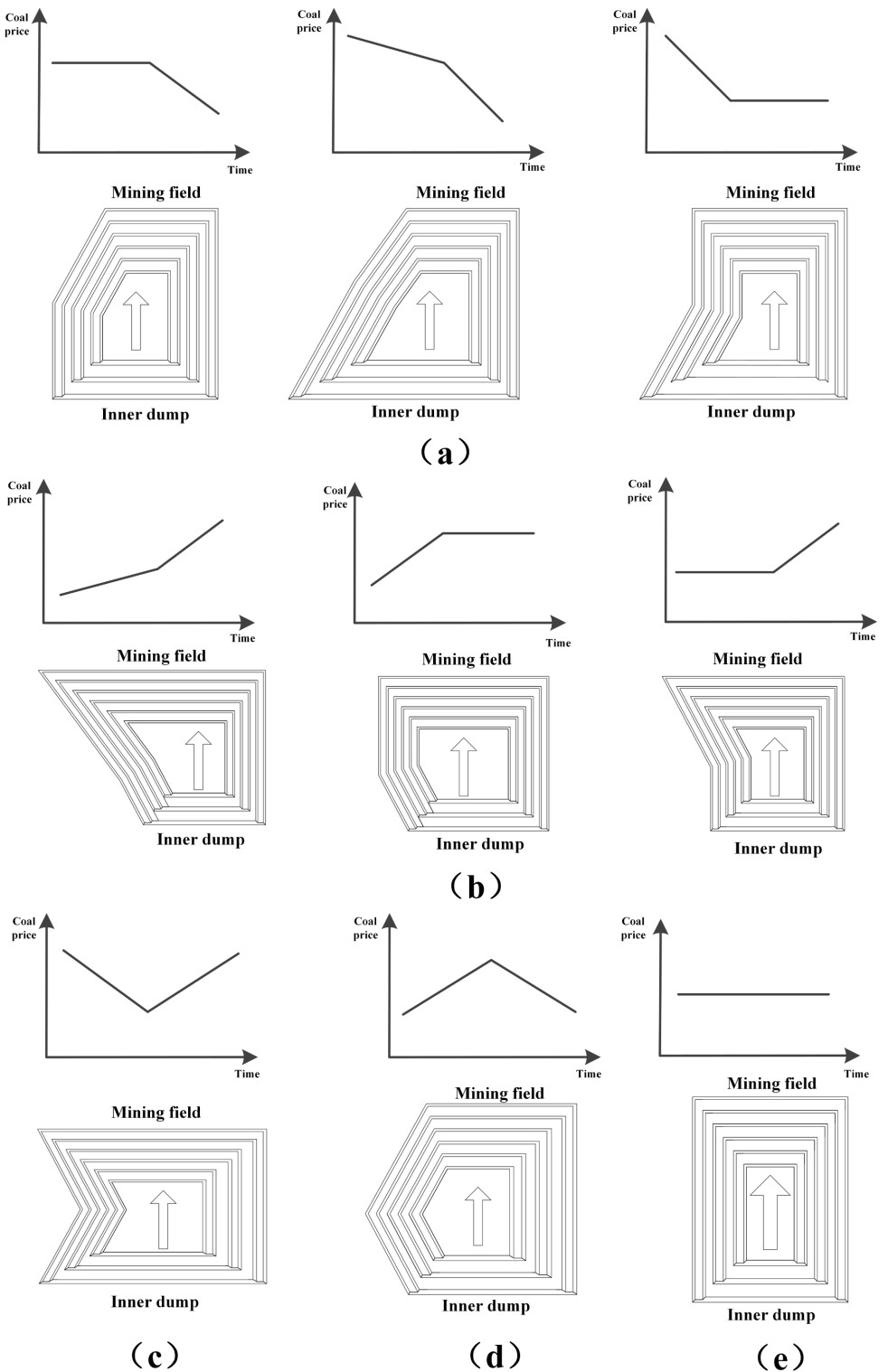

**Fig 8. Schematic diagram of open pit mine boundary fluctuating with coal price.**

aversion coefficient is positive. This method of measuring risk is known as entropy risk measurement. When $\alpha < 0$, raise the open-pit mine's boundary to increase revenue; when $\alpha > 0$, shrink the boundary to decrease the loss of coal that is stripped from the mine. Select the previous coal price fluctuation cycle as the object of this study, according to the topology to select the minimum open-pit mine optimization design in this period $c_s = 80\%$, that is, the adjustment range of the working line is no more than 20% of the existing length. The coal price shows a combined change trend, when the preset raw coal price is in the range of 0.8–1.2 times the actual coal price, it can meet the requirement of reaching degree in this period; when it exceeds this range, it can not meet the demand of achieving degree in this period due to the influence of mining conditions.

That is, the boundary adjustment should closely track changes in coal prices, but taking into account the size of the adjustment, the working line should be made as close as possible to the length of the working line determined by the actual coal price, and potential risks to the open-pit mine should be managed. The risk that an open-pit mine faces in the event of fluctuating coal prices is measured using entropy risk: (1) When $\alpha = 0$, it denotes risk neutrality, meaning that the boundary does not need to be adjusted at this time because there is no difference between the preset coal price index and the actual coal price. (2) When the $\alpha$ value is +0.1 and +0.2, it indicates that the company must assume the risk of loss since the open-pit mine's preset coal price is higher than the actual coal price. The amount of the reduction in the open-pit mine's boundary, or the desire for the level of achievement, depends on the difference between coal prices at this time. (3) The preset coal price of the open-pit mine is lower than the actual coal price when the $\alpha$ value is -0.1 to -0.2, therefore it is required to take into account the demand of expanding the open-pit mine border to attain the degree of maturity.

It is clear that the boundary expansion must be taken into account for the inclined coal seam open-pit mine, which is impacted by the occurrence state of the coal seam, and that the expansion realm presents a greater challenge than the narrowing realm. Therefore, the realm will only be expanded when the price of coal is consistently higher than the predetermined coal price. In other words, the amplitude and frequency of boundary contraction are much higher than those of border expansion for the final boundary plan.

## 5. Results

### 5.1 Coal price prediction model construction and extrapolation prediction

With the help of the seven steps of the ARIMA model described above, the annual forecast of coal prices is carried out. The first-order difference series is a stationary time series, while the original data is a non-time stationary series, according to the stationarity analysis. The unit root check value of the first-order differential data is-12.1012, which is less than 1%, 5% and 10%, and the $p$ value is 0, less than 0.05. it can be seen that the series accords with the standard of stationary time series. Autocorrelation diagram (ACF) and partial autocorrelation diagram (PACF) are made for the first-order differential time series of coal price, and the results are shown in Fig 9.

It can be seen from Fig 9 that the autocorrelation diagram oscillates obviously after order 1, and the partial auto-correlation figure converges to 0 after order 2, and the model parameter is selected as ARIMA (2,0,1). The residual and white noise tests of the model show that the residual sequence is a white noise sequence. It shows that the ARIMA model is suitable for prediction. Fig 10 displays the ARIMA model's projected outcomes.

As shown in Fig 10, through the comparative analysis of the relative error between the predicted value and the actual value of coal price from October 2018 to June 2020, the absolute value of relative error is less than 5%, and the predicted raw coal price has a high degree of fit,

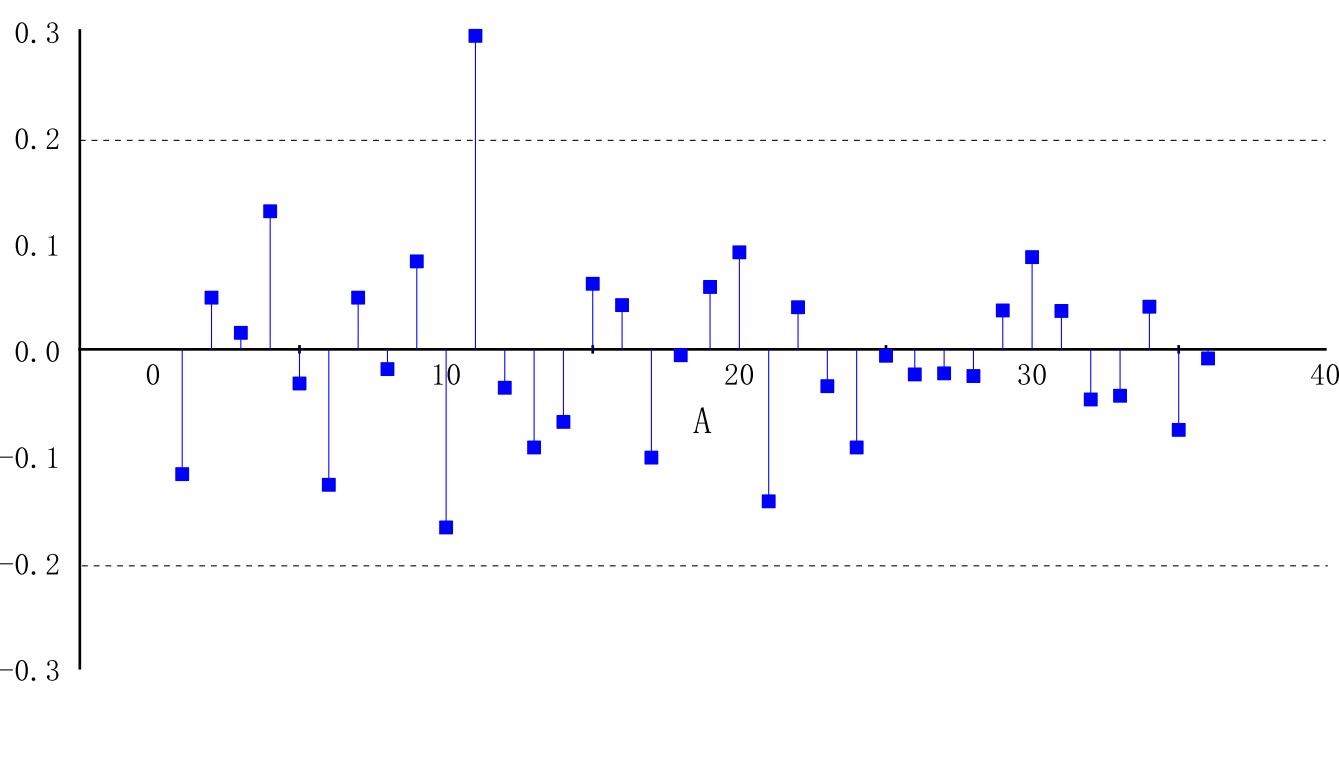

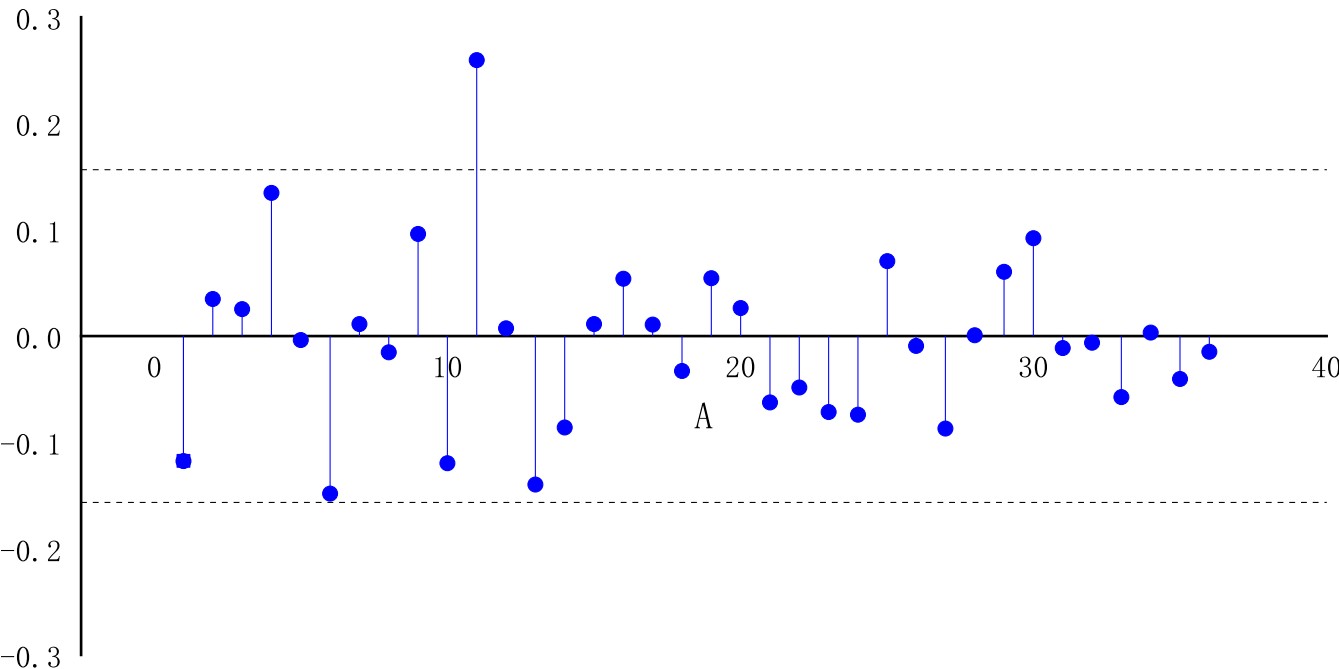

**Fig 9. The ACF and PACF diagrams of coal price time series.** (a) autocorrelation graphs of first-order difference sequences. (b) partial autocorrelation graphs of first-order difference sequences.

which can provide a theoretical basis for the adjustment of the future realm. On this basis, the coal price data are extrapolated to 2021 and 2022, which shows a downward trend of coal price. within this range, the mining boundary of open-pit mine should be reduced in time and the loss should be reduced.

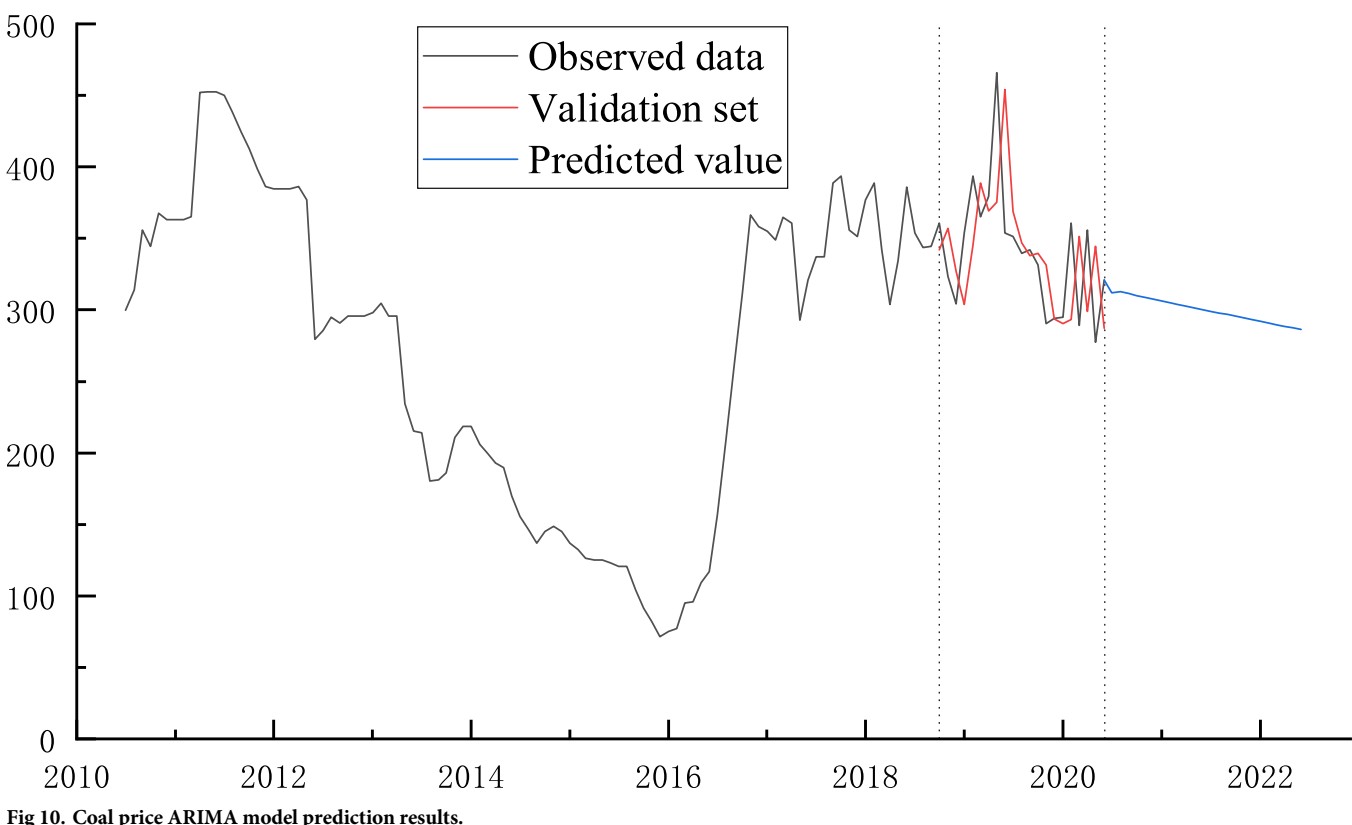

**Fig 10. Coal price ARIMA model prediction results.**

## 5.2 Dynamic adjustment of state under the fluctuation of coal price

Since the fixed coal price index cannot be adjusted to account for changes in coal price, it was chosen as the basis for drawing the original border of an open-pit mine. The enterprise typically bears more risk the larger the discrepancy between the predetermined coal price index and the actual coal price data.

By extrapolating the data of coal price in the range of 2021 ~ 2022, we can see that there is a downward trend in coal price. Given that the deviation value of coal price is kept within 20% at a certain time, that is, the minimum degree of achievement is 80%, Fig 11 depicts the correlation between the shifting of the mining boundaries and the shifting coal price.

The price of raw coal has gone up and down twice, with an average price of 285.85 yuan and a significant variation in price, according to the shifting pattern of the coal price series. The open-pit mine's final mining boundary is dynamically changed in accordance with the price trend for coal. In Fig 11(B), the adjustment diagram is displayed. The risk brought on by the preset profit index diverging from the actual situation can be successfully dealt with by altering the border, which will also increase the enterprise's advantage.

## 5.3 Analysis and evaluation of two-dimensional boundary adjustment scheme

From 2010 to 2022, it is presumable that the small open-pit coal mine will be in a phase of partial internal dis-charge, during which only a small portion of the stripped material will be transferred to the outer dump and the majority to the inner dump for disposal. The remainder of the flat plate exfoliation material is released internally, and it is assumed for the sake of

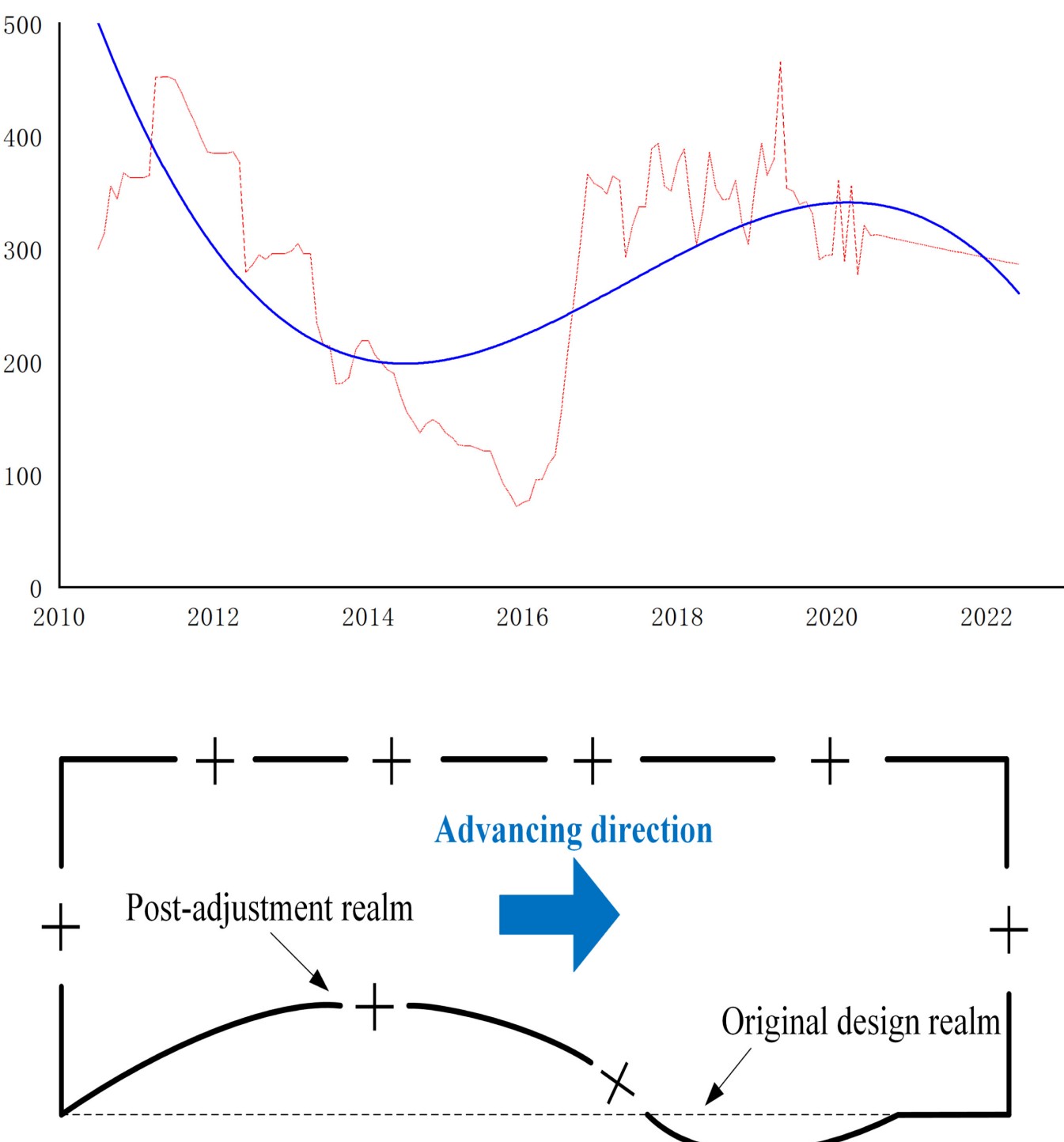

**Fig 11. Comparison of final state adjustment of open pit mines and coal price volatility.** (a) Fluctuation trend of coal price time series. (b) The trend of boundary change in the corresponding period of open-pit mine.

simplification that the final flat plate stripping material is transferred to the outer dump. The inner and outer dumps, as well as the coal bunker, are the three destinations in the mining bor-der block.

As seen in Fig 12, the "1," "2," and "3" blocks are delivered to the coal bunker, the inner dump, and the outer dump, respectively, assuming that each block has a volume of 1×1×1 m. According to the correlation between the selling price of raw coal and the predetermined coal price, the profits in various mining boundaries will be explained be-low. The following are the main findings:

(1) When the actual coal price and the preset coal price are equal, there is no difference between the two prices, making the situation risk-neutral ($\alpha = 0$). The mine's net profit, as cal-culated using Eq (4), is 10 yuan.

(2) Only when the actual coal price reaches 20% of the preset coal price, or 80% success, is the actual coal price higher than the preset coal price. The net profit in the pit is determined using Eq (5) when the price of raw coal in the limit state is 67.6 yuan.

(3) When the actual coal price is less than the preset coal price, the value hits 90% at which point the border can be lowered. This occurs when the actual coal price is 10% less than the preset coal price. Eq (5) is used to deter-mine the pit's net profit when the price of raw coal in the limit condition is 56.8 yuan.

It can be observed that altering the open-pit mine's boundary can raise the net profit in the pit by around 104% and 20%, respectively, while ensuring that the risk can be controlled. This occurs when the price of raw coal grows by 20% and lowers by 10%.

## 5.4 Test of the effect of dynamic adjustment of the boundary during the forecast period of coal price

As mentioned above, the production process of open-pit mine has obvious periodicity, and the threshold can be adjusted by different boundaries according to the construction and develop-ment period of open-pit mine, that is, the value of $\alpha$ is a dynamic process. For the sake of con-venient description, this paper assumes that the small open-pit mine is in the period of operation, that is, the length of the working line does not change greatly, and the adjustment effect of the whole life cycle will be further illustrated by an example of a large open-pit coal mine in the next section.

The boundary adjustment time is set to 12 years using the fluctuation trend of coal price in Fig 11 as the foundation. Table 2 displays the contrast between the actual coal price and the predetermined coal price for each year throughout the adjustment period.

As shown in Table 2, the open-pit mine boundary must be altered in time to satisfy the requirement for risk minimization when the degree of achievement in a particular period of the adjustment period cannot fulfill the demand.

## 5.5 Evaluation system of attainment degree of two stages in coal price forecast period

As can be seen from the description in the previous section, there is a demand for dynamic changes in the boundary of open-pit mines, which is a unified standard, and the minimum degree of achievement of each stage is 80%, that is, the boundary will be adjusted only when the degree of achievement of a certain stage is less than 80%. At the same time, taking into account the limitations of mining procedures, the degree of adjustment within a specific period of time is not more than 20%. Fig 13 shows the comparative relationship between the coal price and the actual raw coal price in the process of using section 2.3 multi-step boundary optimization method. Among them: the demand range of attainment degree in different

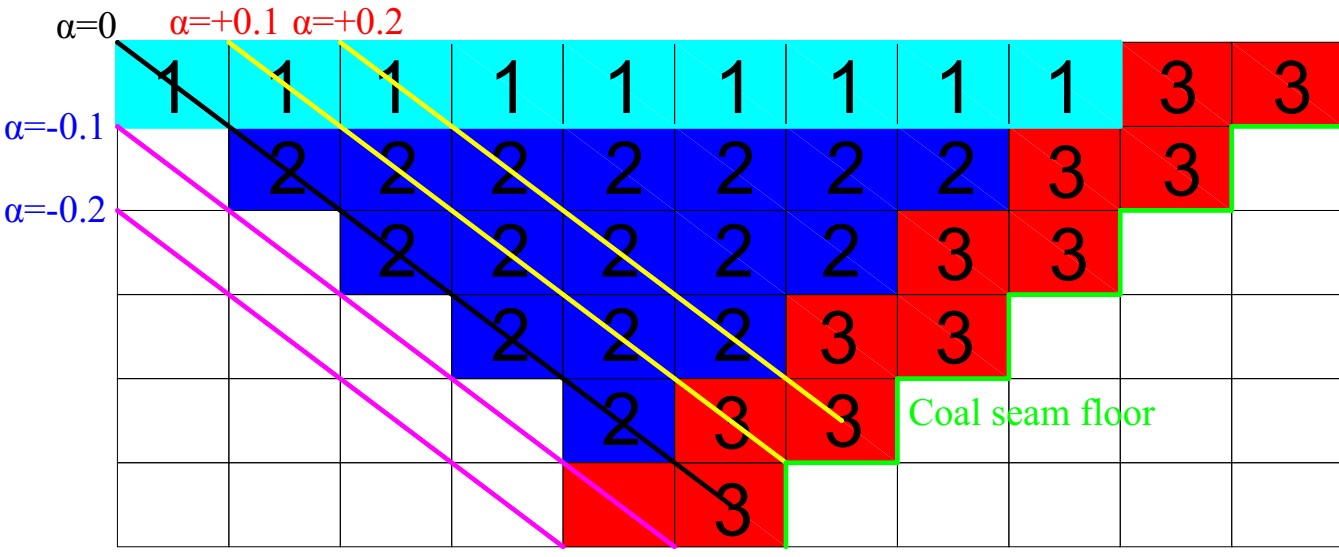

**Fig 12. Schematic diagram of the block model section of a small open-pit mine.**

production stages is 0.8–1.2 times of the actual coal price; the use of coal price in each stage well fits the changing trend of actual coal price, and the use of coal price in most stages is in the relative demand range.

As shown in Fig 13 under the condition of meeting the mining constraints, the sliding window method is used to achieve the maximum value in two stages, combined Eq (5). In 11 production stages, the degree of achievement is 10% as the adjustment interval, and only one set of optimal nesting pits is $s_1^{1.2} \rightarrow s_2^1 \rightarrow s_3^{0.9} \rightarrow s_4^{0.8} \rightarrow s_5^{0.8} \rightarrow s_6^{0.8} \rightarrow s_7^{1.2} \rightarrow s_8^{1.2} \rightarrow s_9^{1.2} \rightarrow s_{10}^1 \rightarrow s_{11}^1$. When the annual push progress of the open-pit mine is a fixed value, the overall working line length showed a trend of change, and the net present value increased by 104%, 104%, 41.6%, 27.2%, 61.8%, 89.4%, 67.3%, 40.7%, 8.9%, 8.9% and 8.9% respectively compared with that before adjustment. The average net present value during the forecast period is 51.15%. Under the premise of ensuring economic benefits, this method can minimize enterprise risk and take

**Table 2. Actual and predetermined coal prices were compared during the adjustment period.**

| Years | Actual coal price (yuan) | Preset coal price (yuan) | Differentials | achievement degree $c_s$ | Whether to adjust the realm |
|---|---|---|---|---|---|
| 2010 | 340.83 | 285.85 | 54.98 | 80.77% | No |
| 2011 | 413.23 | | 127.38 | 55.44% | Expansion |
| 2012 | 329.59 | | 43.74 | 84.70% | No |
| 2013 | 236.19 | | -49.66 | 82.63% | No |
| 2014 | 171.23 | | -114.62 | 59.90% | Reduction |
| 2015 | 113.29 | | -172.56 | 39.63% | Reduction |
| 2016 | 186.16 | | -99.69 | 65.13% | Reduction |
| 2017 | 350.57 | | 64.72 | 77.36% | Expansion |
| 2018 | 346.78 | | 60.93 | 78.68% | Expansion |
| 2019 | 354.98 | | 69.13 | 75.82% | Expansion |
| 2020 | 313.47 | | 27.62 | 90.34% | No |
| 2021 | 299.74 | | 13.89 | 95.14% | No |
| 2022 | 289.12 | | 3.27 | 98.86% | No |

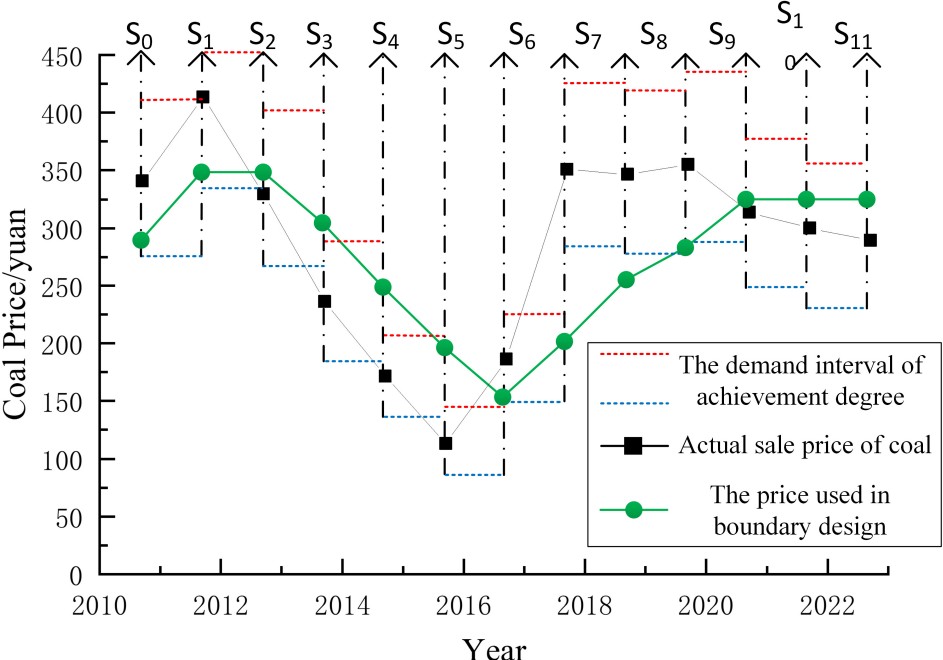

**Fig 13. Comparison of the multi-step boundary optimization design process's used and real coal prices.**

into account the needs of profit and risk management and control. However, it should be noted that in actual production, due to the change of the boundary, the mining cost may change. The average net present value improved by the dynamic optimization method of the boundary in this paper may not reach the proportion in the example, but it is certain that the net present value can be improved under the premise of ensuring the controllable risk.

## 6. Discussion

In this article, considering that in the three-dimensional nested method often used in the open-pit mine boundary optimization design, the block preset economic value is a fixed value, resulting in a relatively fixed boundary, it is difficult to effectively cope with the changes in the economic environment, leading to the risk of enterprises. By introducing uncertainty research into the boundary optimization method, a two-stage boundary dynamic optimization method considering risk nested attributes is proposed. This method uses the entropy risk measurement method to quantify the economic risk, and uses the "achievement degree" index proposed in this paper to measure the dynamic optimization effect of the boundary. The uncertainty of economic indicators is introduced into the block model, and finally the medium and long-term boundary optimization problem is expressed as a two-stage mixed integer problem.

The coal price data from 2010 to 2020 are collected, and the trend of coal prices is predicted by means of econometric methods. The prediction results are applied to the adjustment of the realm, so as to effectively deal with the risks caused by the deviation of the preset profit index from the actual scene and improve the efficiency of the enterprise. Taking an idealized small inclined coal seam open-pit mine as an example, the boundary adjustment strategy under coal price fluctuation is given. The results show that by adjusting the medium and long-term boundary according to the changing trend of coal price, the ' optimization design achievement degree ' in most periods can meet the preset demand, and finally, a set of optimal mining

nested pits can be obtained, which further improves the enterprise benefit under the condition of risk control.

The reason why this paper is important is that it proposes a method to optimize open-pit mining projects by considering changes in economic indicators, which can improve the net present value of coal enterprises while controlling risks. The application example of small inclined coal seam open-pit mine shows that by tracking the change trend of coal price, the average net present value of the proposed method is increased by 51.15% during the 11-year coal price prediction period.

However, due to the limitations of current research methods and means, there are still many shortcomings in this paper, which need to be further studied. (1) Only the change of eco-nomic indicators is considered, and the research on the inherent uncertainty of the block model (such as geological uncertainty) is lacking; (2) The traditional ARIMA model is used to predict the coal price, and the popular group decision optimization algorithm will help to obtain more accurate results; (3) The selection of risk aversion coefficient depends on past experience, but due to the small amount of data, there are some doubts about its credibility. (4) Only the influence of coal price change on the boundary is considered, and other factors with less influence are ignored. The research on the dynamic optimization of the boundary under the condition of multi-factor coupling is insufficient. In general, the method proposed in this paper can improve the efficiency of enterprises as much as possible under the condition of controllable risk, which has a positive effect on the equation of medium and long-term boundary adjustment strategies of open-pit mines.

## 7. Conclusion and future works

In this article, by analyzing the shortcomings of the existing three-dimensional block nesting method, the uncertainty research is introduced into the open-pit mine boundary optimization design. On the basis of considering the characteristics of open-pit mining, a dynamic optimization evaluation system is constructed. By predicting the coal price data, a medium and long-term boundary dynamic optimization method suitable for actual production is proposed, and the effectiveness of the proposed method is verified by using the idealized two-dimensional block. The main conclusions are as follows:

(1) by combining the entropy risk measurement with the nesting idea of opencast mine boundary optimization, the uncertainty is brought into the optimization decision-making link. In addition to the original goal of maximum net present value, multi-stage nested pits are obtained by considering risk minimization, taking into account both eco-nomic benefits and enterprise risks.

(2) according to the production characteristics of open-pit mine and considering the division of the whole life cycle, the concept of "optimal design achievement degree of open-pit mine" is put forward. Given different risk levels, the demand of achievement degree in each period is calculated indirectly, and the adjustment of realm is regulated.

(3) by predicting the fluctuation trend of coal price, taking a small open-pit mine with inclined coal seam as an example, taking the demand of "reaching degree" in each stage as a constraint, the profit within the boundary is in-creased by 51.15%, which verifies the effectiveness of the method proposed in this paper.

This article innovatively puts forward the index of "achievement degree" to measure the effect of boundary optimization, and then constructs the evaluation method of open-pit mine boundary optimization. It breaks the tradition of using fixed boundary index as the design standard of boundary optimization in the past, and takes the two requirements of controllable economic risk and maximum economic benefit as the objective function to solve a set of

medium and long-term boundary schemes under different risk aversion coefficients. However, due to the lack of current research methods and means, the next research work will focus on the following three aspects: (1) Develop a dynamic prediction method of economic indicators with faster speed and better accuracy, and improve the feasibility of basic data. (2) Extensive collection of actual field data, improve the risk aversion coefficient selection process, and strive to propose an automated calculation method of time-interval coefficient. (3) Incorporate the production cost, policy constraints and other factors into the boundary optimization method, and develop a multi-factor boundary dynamic optimization design method.

## Author Contributions

**Conceptualization:** Bo Cao.

**Project administration:** Runcai Bai.

**Resources:** Runcai Bai.

**Software:** Guangwei Liu.

**Visualization:** Guangwei Liu.

**Writing – original draft:** Shuai Wang.

**Writing – review & editing:** Shuai Wang, Bo Cao, Runcai Bai, Guangwei Liu.

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
