## [Decision Letter · Decision Letter 0]

25 Oct 2023

PONE-D-23-22464Mid-long term boundary optimization for open-pit mines considering the fluctuation of coal pricePLOS ONE

Dear Dr. Cao,

Thank you for submitting your manuscript to PLOS ONE. After careful consideration, we feel that it has merit but does not fully meet PLOS ONE’s publication criteria as it currently stands. Therefore, we invite you to submit a revised version of the manuscript that addresses the points raised during the review process.

We look forward to receiving your revised manuscript.

Kind regards,

Fausto Cavallaro, PhD

Academic Editor

PLOS ONE

Journal Requirements:

“The project was supported by the National Natural Science Foundation of China (51974144), the 'Jie Bang Gua Shuai' (Take the Lead) of the Key Scien-tific and Technological Project for Liaoning Prov-ince [2021JH1/10400011], and the Discipline In-novation Team of Liaoning Technical University (LNTU20TD-07).”

“The project was supported by the National Natural Science Foundation of China (51974144), the 'Jie Bang Gua Shuai' (Take the Lead) of the Key Scientific and Technological Project for Liaoning Province [2021JH1/10400011], and the Discipline Innovation Team of Liaoning Technical University (LNTU20TD-07).”

“The project was supported by the National Natural Science Foundation of China (51974144), the 'Jie Bang Gua Shuai' (Take the Lead) of the Key Scien-tific and Technological Project for Liaoning Prov-ince [2021JH1/10400011], and the Discipline In-novation Team of Liaoning Technical University (LNTU20TD-07).”

6. Please remove your figures from within your manuscript file, leaving only the individual TIFF/EPS image files, uploaded separately. These will be automatically included in the reviewers’ PDF.

Reviewers' comments:

Reviewer's Responses to Questions

**Comments to the Author**

1. Is the manuscript technically sound, and do the data support the conclusions?

Reviewer #1: Yes

Reviewer #2: No

2. Has the statistical analysis been performed appropriately and rigorously? 

Reviewer #1: Yes

Reviewer #2: N/A

3. Have the authors made all data underlying the findings in their manuscript fully available?

Reviewer #1: Yes

Reviewer #2: Yes

4. Is the manuscript presented in an intelligible fashion and written in standard English?

Reviewer #1: Yes

Reviewer #2: Yes

5. Review Comments to the Author

Reviewer #1: This article takes boundary design as its starting point and, recognising the need to transcend the influence of coal prices, introduces the concept of "degree of achievement" to articulate the correlation between optimisation outcomes and metric variations. The paper is comprehensive in its content, well-structured, and is recommended for acceptance with some suggested modifications. However, there are certain areas in the paper that require revision. The following issues are listed for your consideration:

1. In the abstract, the author mentions, "this article proposes the concept of 'achievement degree' to reflect the correlation between optimization results and changes in indicators. Taking the coal price index as an example, the risk resistance index is calculated in reverse through the degree of achievement." However, in practical applications, obtaining data related to coal prices is subject to a lag, which raises concerns regarding the calculation of the "achievement degree" using this approach.

2. In Section 2.1, the author mentions, "To ensure the continuity of the project, a phased or partitioned approach is generally used to divide the final mining boundary into several smaller mining areas for mining." However, in reality, a phased or partitioned approach is typically employed only when the mining scope of an open-pit mine is sufficiently large. As far as I am aware, there are numerous small-scale open-pit mines worldwide with annual production capacities in the range of only a few hundred thousand tons. Therefore, the statement may be inaccurate and should be revised accordingly.

3. The author divides the boundaries into final mining boundaries, medium-term boundaries, and short-term boundaries based on the length of their service time. However, there is some confusion in the description, particularly regarding the final mining boundaries, which are determined based on mining rights. This aspect needs to be clarified in greater detail for better understanding.

4. Similarly, in Figure 2(a), when explaining boundary adjustment measures, the author suggests that the boundaries of near-horizontal coal seam open-pit mines can be expanded. However, in reality, boundary expansion may not be feasible due to constraints imposed by mining rights. This point should be revised to accurately reflect the practical limitations on boundary expansion.

5. In the text explaining Figure 2(b), the author suggests that mining depth can vary, and economic indicators can change. However, it's important to note that changes in economic indicators do not necessarily lead to changes in mining depth. There may not be a direct and deterministic relationship between these two factors. This point should be clarified to avoid any misinterpretation.

6. In Section 2.2 of the paper, the author discusses the "achievement degree" and calculates it using actual values compared to preset values. However, obtaining actual values can be challenging, making the "achievement degree" more of a predictive value. Predicting the "achievement degree" accurately for different time periods is indeed a challenging task. It appears that this section's description may be insufficient and should address the complexities and uncertainties involved in predicting the "achievement degree."

7. In Section 3.3, the author adjusts boundaries by changing the working length. However, the example in the paper mainly focuses on inclined coal seams. It is suggested to enhance the effectiveness of the examples by including more detailed explanations and illustrations specifically related to boundary adjustments for inclined coal seam open-pit mines. This would provide a more comprehensive and representative illustration of the concept.

8. In Section 5.3 of the paper, there is a lack of explanation regarding the calculation of the α value. It is recommended to provide supplementary information on the calculation of the α value, including its definition, calculation method, relevant parameters, or formulas.

9. In the conclusion of the paper, it is stated that the proposed method can increase profits by approximately 50%, but it lacks corresponding supporting evidence.

10. The text in Figure 2 is of small size and presents aesthetic concerns. Figure 3 exhibits some monotony and should be enhanced in terms of quality. The curves in Figure 9(a) lack clarity. Figure 12 contains scrambled text, which may be attributed to font or encoding problems. Table 2 suffers from formatting issues.

Reviewer #2: This paper discusses how to improve the economic benefits of open-pit mining projects by considering changes in economic indicators. It introduces the concept of "achievement degree" to measure the relationship between optimization results and indicator changes. Using the example of coal prices, it calculates a risk resistance index and adjusts mine boundaries based on predicted price trends. This approach optimizes mining design while managing risk, increasing net profit by around 50% in a case study of a small coal mine. This paper is important because it introduces a method to optimize open-pit mining projects by accounting for changes in economic indicators, ultimately increasing profitability while managing risk. A manuscript has a practical application and also provides important theoretical for the next studies. The paper can be accepted for publication after providing the corrections mentioned below.

Point 1. It is challenging to read the paper. Why haven't the authors used a commonly known IMRaD structure? The study should adhere to a conventional pattern, clearly stating Methods, Results, and Discussion, to ensure that readers can easily grasp the manuscript's essence.

Point 2. In the Introduction section, an enhanced literature review is required. Mining in context circular economy need to be discussed. This aspect needs significant improvement to provide a more balanced perspective.

Point 3. It will be great if the authors show some description in context – Why it is important to conduct this study?

Point 4. What are the limitations of your research?

Point 5. The aim and the tasks must be highlighted at the end of the Introduction section.

Point 6. Detailed consideration of resource management, such as the depletion of mineral resources, could have provided a more comprehensive view of the sustainability of mining operations.

Point 7. The paper does not discuss the legal and regulatory aspects of mining, which can significantly impact the feasibility and design of open-pit mines. Addressing these factors would provide a more holistic view of the challenges and opportunities in mining optimization.

Point 8. To add more information (according to Point 2) please consider the suggested research in your paper when enhancing the introduction section. I believe they are worth considering in your paper. It will show that the problem of heavy metal pollution is important not only in China (enhanced review of non-China authors only are more than welcome).

Koval, V., Kryshtal, H., Udovychenko, V., Soloviova, O., Froter, O., Kokorina, V., & Veretin, L. (2023). Review of mineral resource management in a circular economy infrastructure. Mining of Mineral Deposits, 17(2), 61-70. https://doi.org/10.33271/mining17.02.061

Upadhyay, A., Laing, T., Kumar, V., & Dora, M. (2021). Exploring barriers and drivers to the implementation of circular economy practices in the mining industry. Resources Policy, 72, 102037. https://doi.org/10.1016/j.resourpol.2021.102037

etc…

Point 9. Address e a short description of further research.

Point 10. The novelty of the paper must be highlighted in the conclusions section.

In general, the presented article leaves a positive impression and, after eliminating these comments and taking into account the recommendations made, it can be recommended for publication in the journal “PLOS ONE”.

6. PLOS authors have the option to publish the peer review history of their article (what does this mean?). If published, this will include your full peer review and any attached files.

Reviewer #1: No

Reviewer #2: No

---

## [Author Response · Author response to Decision Letter 0]

7 Nov 2023

Response Letter

Dear Editor, Dear reviewers

We would like to express our gratitude for your time and effort in reviewing the previous version of our manuscript. We appreciate all your comments and suggestions, and we have addressed each of them in our revised submission. Please find our responses to your comments itemized below, along with the revisions/corrections we have made in the resubmitted files. It is worth noting that the authors have modified the structure and layout of the article. We hope this will not cause any trouble to your retrial.

Thanks again!

SUGGESTIONS FROM EDITOR

Comment: Thank you for submitting your manuscript to PLOS ONE. After careful consideration, we feel that it has merit but does not fully meet PLOS ONE’s publication criteria as it currently stands. Therefore, we invite you to submit a revised version of the manuscript that addresses the points raised during the review process. When submitting your revision, we need you to address these additional requirements.

Response to comment: We have comprehensively revised the manuscript according to the reviewer's opinions, and comprehensive changes were made to meet the additional requirements of the journal. We apologize for our oversight and thank you for your efforts on this article.

RESPONSE TO THE REVIEWERS

Reviewer 1

Dear reviewer

Thank you for recognizing our efforts in improving the article. We carefully considered your comments and made revisions to address the issues and errors in the paper. We apologize for any shortcomings and hope that the revised version of the paper can address these concerns.

Comment 1: In the abstract, the author mentions, "this article proposes the concept of 'achievement degree' to reflect the correlation between optimization results and changes in indicators. Taking the coal price index as an example, the risk resistance index is calculated in reverse through the degree of achievement." However, in practical applications, obtaining data related to coal prices is subject to a lag, which raises concerns regarding the calculation of the "achievement degree" using this approach.

Response to comment: In the description of "achievement degree", this paper describes the confusion. As the reviewer said, there is a lag in obtaining data related to coal price, so it is necessary to provide the demand of "achievement degree" in advance on the basis of accurate prediction of coal price and economic analysis method. In this paper, the method of calculating the achievement degree of coal price is only for the consideration of narrative convenience. Therefore, we modify the narration of this part and perfect the narration of "achievement degree". Additional statements are as follows: 

The "achievement degree" at a specific time represents the correlation coefficient between the change rate of economic indicators and the change rate of boundary adjustment indicators, which is used to test the degree of fitting between the optimization results and the predetermined demand, and is expressed by the symbol cs. Its value range is usually between −1 and 1, −1 indicates a completely negative correlation, 1 indicates a completely positive correlation, and 0 indicates no linear correlation. Its calculation formula is , where, CR-economic index change rate, LR-boundary adjustment rate.

Comment 2: In Section 2.1, the author mentions, "To ensure the continuity of the project, a phased or partitioned approach is generally used to divide the final mining boundary into several smaller mining areas for mining." However, in reality, a phased or partitioned approach is typically employed only when the mining scope of an open-pit mine is sufficiently large. As far as I am aware, there are numerous small-scale open-pit mines worldwide with annual production capacities in the range of only a few hundred thousand tons. Therefore, the statement may be inaccurate and should be revised accordingly.

Response to comment: The authors are sorry to have overlooked the lack of zoning or staging problems in small open pit mines. We have revised this statement to "According to the size of resource reserves and annual production capacity, open-pit mines can be divided into small, medium and large open-pit coal mines. Generally speaking, the final mining boundary of large and medium-sized open-pit mines is very wide. In order to ensure the continuity of the project, the final mining boundary will be divided into several smaller mining areas by stages or partitions. Due to the small reserves of resources, small open-pit coal mines are generally divided into only one mining area for mining".

Comment 3: The author divides the boundaries into final mining boundaries, medium-term boundaries, and short-term boundaries based on the length of their service time. However, there is some confusion in the description, particularly regarding the final mining boundaries, which are determined based on mining rights. This aspect needs to be clarified in greater detail for better understanding.

Response to comment: We apologize that the classification method is not the same standard, which leads to the confusion of the description. Therefore, the authors change the description according to the logic of mining planning, in the order of mining right boundary, zoning (stage) boundary, medium and long-term boundary, and short-term boundary. The revised statement is" As mentioned above, the mining boundary is the basic document for the preparation of the mining plan. The mining plan can be divided into overall planning, long-term planning, and annual planning".

Comment 4: Similarly, in Figure 2(a), when explaining boundary adjustment measures, the author suggests that the boundaries of near-horizontal coal seam open-pit mines can be expanded. However, in reality, boundary expansion may not be feasible due to constraints imposed by mining rights. This point should be revised to accurately reflect the practical limitations on boundary expansion.

Response to comment: We apologize for not being thoughtful. As the reviewer said, for the open-pit mine of near horizontal coal seam, because the change of mining conditions in the whole mining range is not very obvious, it is not feasible in policy to adjust the boundary at will due to the restriction of mining rights. Therefore, we have revised this part of the description, limiting the boundary adjustment of horizontal open-pit mine to the zoning boundary, and the revised statement is as follows: As shown in figure 2 (a), the adjustment of the open-pit mining boundary of the near-horizontal coal seam can be approximately regarded as caused by the outward expansion or inward contraction of the surface boundary. The final mining boundary is generally the mining right boundary, which can only be adjusted when there are major changes in the internal and external environment. The boundary line of the partition boundary can change with the change of economic indicators, so the research object of the dynamic adjustment of the boundary is only for the partition boundary.

As shown in figure 2 (b), due to the inclined coal seam in open-pit mine, the change of economic index affects the economy of coal resource mining in the original boundary, which leads to the change of optimal mining depth. Therefore, for the inclined coal seam open-pit mine, the dynamic adjustment of the boundary is not only applicable to the partition boundary, but also to the final mining boundary.

At the same time, we have modified Figure 2 (Re-encoded Figure 4) to add a representation of the mining domain. 

Comment 5: In the text explaining Figure 2(b), the author suggests that mining depth can vary, and economic indicators can change. However, it's important to note that changes in economic indicators do not necessarily lead to changes in mining depth. There may not be a direct and deterministic relationship between these two factors. This point should be clarified to avoid any misinterpretation.

Response to comment: As the reviewer said, the change of economic indicators will not necessarily cause the change of mining depth, and there is no inevitable relationship between the two. Therefore, we have made some clarifications to explain the adjustment mechanism of economic indicators on mining depth, and the revised statement is as follows: As shown in figure 2 (b), due to the inclined coal seam in open-pit mine, the change of economic index affects the economy of coal resource mining in the original boundary, which leads to the change of optimal mining depth. Therefore, for the inclined coal seam open-pit mine, the dynamic adjustment of the boundary is not only applicable to the partition boundary, but also to the final mining boundary.

Comment 6: In Section 2.2 of the paper, the author discusses the "achievement degree" and calculates it using actual values compared to preset values. However, obtaining actual values can be challenging, making the "achievement degree" more of a predictive value. Predicting the "achievement degree" accurately for different time periods is indeed a challenging task. It appears that this section's description may be insufficient and should address the complexities and uncertainties involved in predicting the "achievement degree."

Response to comment: We are sorry for the objection, so we modified this part of the narrative. The calculation of the minimum achievement degree is determined by the risk aversion coefficient. In the actual production process, the 'achievement degree' of the next production stage is calculated by predicting the change of economic indicators, so as to compare with the preset minimum achievement degree of the production unit, and then determine whether to adjust the boundary. 

Comment 7: In Section 3.3, the author adjusts boundaries by changing the working length. However, the example in the paper mainly focuses on inclined coal seams. It is suggested to enhance the effectiveness of the examples by including more detailed explanations and illustrations specifically related to boundary adjustments for inclined coal seam open-pit mines. This would provide a more comprehensive and representative illustration of the concept.

Response to comment: Because the benefit of open pit mine in inclined coal seam has more obvious change to economic index, the examples in this paper are mainly concentrated on inclined coal seam. However, the description of this part is short, so the author expands the description of this part as required, and the expanded description is as follows: As shown in Figure 4, the boundary adjustment of inclined coal seam open-pit mine not only involves the adjustment of partition and stage boundary, but also includes the final mining boundary change caused by the change of mining depth. Therefore, it is more representative and convincing to choose inclined coal seam open-pit mine as an example. Thus, an idealized small open-pit coal mine is chosen as the research object in order to demonstrate the usefulness of the method in this work in solving the boundary optimization problem of open-pit mine under unknown conditions.

Comment 8: In Section 5.3 of the paper, there is a lack of explanation regarding the calculation of the α value. It is recommended to provide supplementary information on the calculation of the α value, including its definition, calculation method, relevant parameters, or formulas.

Response to comment: Similar to the first comment, the author made a mistake in the statement of the "degree of achievement", so in this section we have added information about the calculation of α values, including definitions, calculation formulas, and other descriptions. The following is the content of the article added.

In entropy risk measurement, the selection of risk aversion coefficient α is very important. If α is positive, it will increase the weight of high probability events in the risk measure, thereby increasing the overall risk. If α is negative, it will reduce the weight of high probability events, thereby reducing the overall risk. The determination of α value usually needs to consider the specific decision-making situation and the preference of decision-makers. Based on the sensitivity analysis method to determine the sensitivity of α value to decision-making, the appropriate value is selected according to the historical data information and the preference of decision-makers. However, the decision-making problem and the preference of decision makers may change with time, and the α value needs to be updated and corrected according to the actual situation.

Comment 9: In the conclusion of the paper, it is stated that the proposed method can increase profits by approximately 50%, but it lacks corresponding supporting evidence.

Response to comment: Your opinions are very valuable. Due to the lack of calculation basis, the credibility of the data is doubtful. Therefore, we have added the content about profit calculation, hoping to improve the credibility of the data. The additions are as follows: When the annual push progress of the open-pit mine is a fixed value, the overall working line length showed a trend of change, and the net present value increased by 104%, 104%, 41.6%, 27.2%, 61.8%, 89.4%, 67.3%, 40.7%, 8.9%, 8.9% and 8.9% respectively compared with that before adjustment. The average net present value during the forecast period is 51.15%.

Comment 10: The text in Figure 2 is of small size and presents aesthetic concerns. Figure 3 exhibits some monotony and should be enhanced in terms of quality. The curves in Figure 9(a) lack clarity. Figure 12 contains scrambled text, which may be attributed to font or encoding problems. Table 2 suffers from formatting issues.

Response to comment: We are very sorry for the format problem of the chart due to our negligence. We have modified the problem you raised and checked all the charts. We hope it will not cause you trouble anymore.

Reviewer 2

Dear reviewer

Thank you very much for taking the time out of your busy schedule to review the manuscript. We have carefully read your comments and fully agree with them. We have addressed each comment one by one, and added content on circular economy, resource management, and mining law and regulation. Thank you again for your efforts and excellent advice.

Thanks again!

Comment 1: It is challenging to read the paper. Why haven't the authors used a commonly known IMRaD structure? The study should adhere to a conventional pattern, clearly stating Methods, Results, and Discussion, to ensure that readers can easily grasp the manuscript's essence.

Response to comment: Sorry for the confusion caused by structural problems. Therefore, we have adjusted the structure of the article to the common IMRaD structure, and at the same time adjusted the proportion of the length of each part of the article, hoping that the revised article can solve the above problems.

Comment 2: In the Introduction section, an enhanced literature review is required. Mining in context circular economy need to be discussed. This aspect needs significant improvement to provide a more balanced perspective.

Response to comment: Thank you very much for your suggestion. As the reviewer said, mining under the background of circular economy is the main trend of future development, but this part seems to have been neglected in the paper. Therefore, we have added the analysis of factors related to circular economy, resource management and regulations and policies in the introduction, hoping to provide a more balanced perspective. The following is the content of the article added.

The activities of mining companies are not only sensitive to market factors (price fluctuations, demand changes), but also sensitive to non-economic global challenges (climate change, regional conflicts). They must cope with the decline in the quality and availability of mineral resources and the increase in the amount of mine waste [19]. Therefore, in the past few years, the issue of sustainable development (SD) has become the subject of many discussions [20]. Mining can have strong negative impacts on the natural environment in the form of soil disturbance, waste rock disposal, water pollution and drainage, landscape destruction, harmful discharges, water contamination by industrial effluents, flora and fauna degradation and other negative impacts on ecosystems [21]. Based on this, the concept of circularity has been gradually introduced into mining [22]. Circular economy describes production in a circular model where markets, regulations, and industrial systems are optimized to design high-performance products, minimize impact, restore or regenerate the environment, and optimize material use [23]. And the mining circular economy refers to the economic system that follows the characteristics and natural ecological laws of mineral resources and mineral products, and takes the efficient development and comprehensive utilization of mineral resources as the core [24]. Most scholars agree that the principle of circular economy should be followed is 'Reduce, Reuse, Recycle', referred to as '3R'. In the process of mining, processing and utilization of mineral resources, the circular economy is mainly manifested as: (1) through mechanization, automation and optimization of mining, to achieve efficient exploitation of resources; (2) By studying the mining and smelting technology of complex and difficult-to-mine and difficult-to-separate ores, the mining dilution rate and ore loss rate are reduced, the recovery rate of beneficiation and smelting is improved, and the total recovery rate of resources is improved. (3) Reduce the discharge of various pollutants such as tailings, coal gangue and mine wastewater, and improve the comprehensive benefits of resource development [25]. Circular economy is one of the key directions of mineral resources conservation and protection of sustainable development policy [26]. For emerging economies, including Brazil, Chile, China and South Africa, their economic dependence on mining and extractive industries is relatively large, so the research of circular economy is very important [27].

Comment 3: It will be great if the authors show some description in context – Why it is important to conduct this study?

Response to comment: As you suggested, we have described why it is important to conduct this research in the introduction, research basis, and other sections. Specifically: The reason why this paper is important is that it proposes a method to optimize open-pit mining projects by considering changes in economic indicators, which can improve the net present value of coal enterprises while controlling risks. The application example of small inclined coal seam open-pit mine shows that by tracking the change trend of coal price, the average net present value of the proposed method is increased by 51.15 % during the 11-year coal price prediction period.

Comment 4: What are the limitations of your research?

Response to comment: Thank you for your suggestion, and I apologize for the lack of this part in the article. Therefore, we have added the description of the research limitations and the next work in the conclusion and prospect of the paper, hoping that the revised paper can meet the requirements. Additional statements are as follows:

However, due to the limitations of current research methods and means, there are still many shortcomings in this paper, which need to be further studied. (1) Only the change of eco-nomic indicators is considered, and the research on the inherent uncertainty of the block model (such as geological uncertainty) is lacking; (2) The traditional ARIMA model is used to predict the coal price, and the popular group decision optimization algorithm will help to obtain more accurate results; (3) The selection of risk aversion coefficient depends on past experience, but due to the small amount of data, there are some doubts about its credibility. (4) Only the influence of coal price change on the boundary is considered, and other factors with less influence are ignored. The research on the dynamic optimization of the boundary under the condition of multi-factor coupling is insufficient.

Comment 5: The aim and the tasks must be highlighted at the end of the Introduction section.

Response to comment: We have completely revised the introduction to highlight the goals and tasks at the end of the introduction section, and again we apologize for our inadequacies. Additional statements are as follows:

Therefore, based on the actual production of open-pit coal mine, the author tries to put forward a method to optimize the mining boundary of open-pit mine by considering the change of coal price under the condition of fully considering the constraints of mining right and law. The ultimate goal is to improve the net present value in a specific period of time under the condition of risk control. Considering the difference between the near horizontal coal seam open-pit mine and the inclined coal seam open-pit mine in the boundary division work, the optimization effect evaluation index under the unified research standard is proposed, which is called ' achievement degree '. The goal of the study is to propose a dynamic adjustment strategy that comprehensively considers constraints including policy, economy, and mining conditions, and is suitable for the actual production of open-pit coal mines.

Comment 6: Detailed consideration of resource management, such as the depletion of mineral resources, could have provided a more comprehensive view of the sustainability of mining operations.

Response to comment: Thank you for your suggestion. In the introduction and some areas of the article, we have added a description about resource management. The revised article provides a more complete picture of the sustainability of mining operations. Additional statements in Introduction are as follows:

Coal resources are the most important basic energy in China. More than 50% of the energy in electric power, metallurgy, chemical industry, building materials and life industries comes from coal resources [15]. However, a large number of coal resources are continuously developed and utilized, and the mining is not scientific and the utilization is not efficient, which makes the environmental and ecological problems caused by mining emerge in endlessly [16]. The coal industry has long adopted extensive management methods. However, due to the inherent scarcity and limitation of resources, the problems of resources and resource utilization are prominent [17].

At the same time, in the process of determining the risk aversion coefficient, it is possible to assume a part of the risk under the conditions of legal constraints, so as to expand the open-pit mine boundary, extend the service life, and ensure the demand for resource management. The specific contents are as follows :

It is worth noting that due to the long-term extensive mining mode adopted by the coal industry, coupled with the large-scale mining and utilization of coal in recent years, mineral resources have been depleted. Because the initial planning and design tends to have better mining conditions and shallower shallow coal resources, the conditions of coal resources in the subsequent development process become worse. For example, when the boundary of open-pit mining in inclined coal seam is delineated, in order to ensure the economy of its boundary, it is usually considered that when the mining depth exceeds a certain range, the deep resources are mined by underground mining. However, it should be noted that the coal resources adjacent to the open-pit mine have a natural disadvantage in terms of safety compared with the original open-pit mining method. Therefore, it is theoretically feasible to expand the original open-pit mining boundary under the conditions allowed. Therefore, in the process of determining the risk aversion coefficient, it is possible to have a tendency to bear certain risks, thereby increasing the reserves of open-pit mine resources, extending the service life and obtaining more resources as much as possible, and making beneficial contributions to the management of mining resources. Of course, in this process, one of the factors that must be considered is the legal factors and environmental protection restrictions.

Comment 7: The paper does not discuss the legal and regulatory aspects of mining, which can significantly impact the feasibility and design of open-pit mines. Addressing these factors would provide a more holistic view of the challenges and opportunities in mining optimization.

Response to comment: It is true that the legal and regulatory aspects of mining are lacking, so the proposed approach is actually limited. Therefore, we have added the narration about this part, hoping that the revised article will not have the above problems again. Additional statements are as follows:

China 's coal resources reserves are in the forefront of the world, but the per capita share is low. If a complete legal system for mining and mining management is not established, it will seriously affect China 's energy management [18]. The mining right system is the core content of the legal system of mineral resources. The mining right clarifies the scope of mining allowed by law in open-pit coal mines and is a rigid constraint that must be observed in the process of delimitation. The complexity and rigor of mining laws and regulations may have a wide impact on the feasibility and design of open-pit mines. Mining companies must closely comply with legal and regulatory requirements when designing and implementing projects to ensure project feasibility and sustainability while meeting environmental and social expectations.

Comment 8: To add more information (according to Point 2) please consider the suggested research in your paper when enhancing the introduction section. I believe they are worth considering in your paper. It will show that the problem of heavy metal pollution is important not only in China (enhanced review of non-China authors only are more than welcome).

Koval, V., Kryshtal, H., Udovychenko, V., Soloviova, O., Froter, O., Kokorina, V., & Veretin, L. (2023). Review of mineral resource management in a circular economy infrastructure. Mining of Mineral Deposits, 17(2), 61-70. https://doi.org/10.33271/mining17.02.061

Upadhyay, A., Laing, T., Kumar, V., & Dora, M. (2021). Exploring barriers and drivers to the implementation of circular economy practices in the mining industry. Resources Policy, 72, 102037. https://doi.org/10.1016/j.resourpol.2021.102037

etc…

Response to comment: The literature provided by the reviewers is very helpful and an important supplement to the improvement of the article. Based on this, we add the following references, hoping to have a better description of circular economy. Additional literature is as follows:

1.Koval, V., Kryshtal, H., Udovychenko, V., Soloviova, O., Froter, O., Kokorina, V., & Veretin, L. (2023). Review of mineral resource management in a circular economy infrastructure. Mining of Mineral Deposits, 17(2), 61-70. https://doi.org/10.33271/mining17.02.061

2.Upadhyay, A., Laing, T., Kumar, V., & Dora, M. (2021). Exploring barriers and drivers to the implementation of circular economy practices in the mining industry. Resources Policy, 72, 102037. https://doi.org/10.1016/j.resourpol.2021.102037

3. Blinova E, Ponomarenko T, Knysh V. Analyzing the Concept of Corporate Sustainability in the Context of Sustainable Business Development in the Mining Sector with Elements of Circular Economy. Sustainability, 2022, 14(13): 8163.

4. UNEP Circularity Platform. Available online: https://buildingcircularity.org/ (accessed on 30 January 2022).

5.Valverde, J.-M.; Avilés-Palacios, C. Circular Economy as a Catalyst for Progress towards the Sustainable Development Goals: A Positive Relationship between Two Self-Sufficient Variables. Sustainability 2021, 13, 12652.

6. Rybak, J.; Gorbatyuk, S.M.; Bujanovna-Syuryun, K.C.; Khairutdinov, A.M.; Tyulyaeva, Y.S.; Makarov, P.S. Utilization of Mineral Waste: A Method for Expanding the Mineral Resource Base of a Mining and Smelting Company. Metallurgist 2021, 64, 851–861.

7. Mining and Metals and the Circular Economy. ICMM. 2016. Available online: https://www.icmm.com/en-gb/publications/ responsible-sourcing/circular-economy (accessed on 3 June 2022).

8. Zhao Y, Zang L, Li Z, et al. Discussion on the model of mining circular economy. Energy Procedia, 2012, 16: 438-443.

9. Wang YM. China Recycling Economy Development and its mineral resources’ sustainable development. Metal Mine 2005; 1-3.

Comment 9: Address e a short description of further research.

Response to comment: Thank you very much for your comments. Sorry for the lack of a description of further research in the article, so we have added a description of further research at the end of the article.

However, due to the lack of current research methods and means, the next research work will focus on the following three aspects: (1) Develop a dynamic prediction method of economic indicators with faster speed and better accuracy, and improve the feasibility of basic data. (2) Extensive collection of actual field data, improve the risk aversion coefficient selection process, and strive to propose an automated calculation method of time-interval coefficient. (3) Incorporate the production cost, policy constraints and other factors into the boundary optimization method, and develop a multi-factor boundary dynamic optimization design method.

Comment 10: The novelty of the paper must be highlighted in the conclusions section.

Response to comment: Thank you for your suggestion, the conclusion does have the problem that the novelty is not outstanding. Therefore, we have revised the conclusion according to your request, hoping to improve the problem of insufficient novelty. The revised conclusions are as follows:

In this article, by analyzing the shortcomings of the existing three-dimensional block nesting method, the uncertainty research is introduced into the open-pit mine boundary optimization design. On the basis of considering the characteristics of open-pit mining, a dynamic optimization evaluation system is constructed. By predicting the coal price data, a medium and long-term boundary dynamic optimization method suitable for actual production is proposed, and the effectiveness of the proposed method is verified by using the idealized two-dimensional block. The main conclusions are as follows:

(1) by combining the entropy risk measurement with the nesting idea of opencast mine boundary optimization, the uncertainty is brought into the optimization decision-making link. In addition to the original goal of maximum net present value, multi-stage nested pits are obtained by considering risk minimization, taking into account both eco-nomic benefits and enterprise risks.

(2) according to the production characteristics of open-pit mine and considering the division of the whole life cycle, the concept of "optimal design achievement degree of open-pit mine" is put forward. Given different risk levels, the demand of achievement degree in each period is calculated indirectly, and the adjustment of realm is regulated.

(3) by predicting the fluctuation trend of coal price, taking a small open-pit mine with inclined coal seam as an example, taking the demand of "reaching degree" in each stage as a constraint, the profit within the boundary is in-creased by 51.15%, which verifies the effectiveness of the method proposed in this paper.

If you have any questions, please do not hesitate to contact me. We value your feedback and will make necessary modifications promptly. Thank you once again for your valuable contributions.

Thank you and best regards.

Your sincerely,

Bo Cao

College of Mining, Liaoning Technical University, Fuxin, 123000, China

Tel.: +86 15841827800. E-mail address: caobo0418@163.com

---

## [Decision Letter · Decision Letter 1]

21 Dec 2023

Mid-long term boundary dynamic optimization of open-pit coal mine considering coal price fluctuation

PONE-D-23-22464R1

Dear Dr. Cao,

We’re pleased to inform you that your manuscript has been judged scientifically suitable for publication and will be formally accepted for publication once it meets all outstanding technical requirements.

Kind regards,

Fausto Cavallaro, PhD

Academic Editor

PLOS ONE

Additional Editor Comments (optional):

The authors addressed the reviewer's comments. The paper can be accepted

Reviewers' comments:

Reviewer's Responses to Questions

**Comments to the Author**

1. If the authors have adequately addressed your comments raised in a previous round of review and you feel that this manuscript is now acceptable for publication, you may indicate that here to bypass the “Comments to the Author” section, enter your conflict of interest statement in the “Confidential to Editor” section, and submit your "Accept" recommendation.

Reviewer #1: All comments have been addressed

2. Is the manuscript technically sound, and do the data support the conclusions?

Reviewer #1: Yes

3. Has the statistical analysis been performed appropriately and rigorously? 

Reviewer #1: Yes

4. Have the authors made all data underlying the findings in their manuscript fully available?

Reviewer #1: Yes

5. Is the manuscript presented in an intelligible fashion and written in standard English?

Reviewer #1: Yes

6. Review Comments to the Author

Reviewer #1: The author made detailed revisions to the paper based on the comments of the reviewers. The quality of the paper has been significantly improved.

7. PLOS authors have the option to publish the peer review history of their article (what does this mean?). If published, this will include your full peer review and any attached files.

Reviewer #1: No

---

## [Editor Report · Acceptance letter]

4 Jan 2024

PONE-D-23-22464R1 

PLOS ONE

Dear Dr. Cao, 

I'm pleased to inform you that your manuscript has been deemed suitable for publication in PLOS ONE. Congratulations! Your manuscript is now being handed over to our production team.

Kind regards, 

on behalf of

Professor Fausto Cavallaro 

Academic Editor

PLOS ONE